# Resource-Aware Federated Self-Supervised Learning with Global Class Representations

**Mingyi Li**
Shandong University

**Xiao Zhang**[*]
Shandong University

**Qi Wang**
Shandong University

**Tengfei Liu**
Hong Kong University of
Science and Technology

**Ruofan Wu**
Coupang

**Weiqiang Wang**
Shanghai Jiaotong University

**Fuzhen Zhuang**
Institute of Artificial Intelligence,
Beihang University
Zhongguancun Laboratory

**Hui Xiong**
Thrust of AI, HKUST(Guangzhou)
Dep. of Com. Sci. and Eng.,
HKUST

**Dongxiao Yu**[*]
Shandong University

## Abstract

Due to the heterogeneous architectures and class skew, the global representation models training in resource-adaptive federated self-supervised learning face with tricky challenges: *deviated representation abilities* and *inconsistent representation spaces*. In this work, we are the first to propose a multi-teacher knowledge distillation framework, namely *FedMKD*, to learn global representations with whole class knowledge from heterogeneous clients even under extreme class skew. Firstly, the adaptive knowledge integration mechanism is designed to learn better representations from all heterogeneous models with deviated representation abilities. Then the weighted combination of the self-supervised loss and the distillation loss can support the global model to encode all classes from clients into a unified space. Besides, the global knowledge anchored alignment module can make the local representation spaces close to the global spaces, which further improves the representation abilities of local ones. Finally, extensive experiments conducted on two datasets demonstrate the effectiveness of *FedMKD* which outperforms state-of-the-art baselines 4.78% under linear evaluation on average.

## 1 Introduction

The federated self-supervised learning (Fed-SSL) has emerged as a highly promising paradigm due to the extremely limited labeled data in real-world scenarios [4, 25, 30]. The Fed-SSL mechanism can learn common representations collaboratively across all the clients without labeled data [7, 8], which could enable the aggregation of knowledge from diverse unlabeled data sources and overcome the limitations caused by the high cost and scarcity of labeled data [24, 28, 31].

Traditional Fed-SSL methods usually assume that each client should train the identical architecture model, such as FedU [33], FedEMA [34], FedCA [30]. But it would not be easy in resource-limited scenarios, especially for the existing arsing large-scale models [1, 26]. As shown in Fig.1, *client A*, *client B* and *client C* might train heterogeneous representation models due to the varying system resources. In addition, real-world data often exhibits skewed class distributions across clients.

---

[*]Xiao Zhang and Dongxiao Yu are corresponding authors. Email:{xiaozhang, dxyu}@sdu.edu.cn

38th Conference on Neural Information Processing Systems (NeurIPS 2024).

Therefore, how to learn global class representations under the heterogeneous architectures and class skew in resource-aware Fed-SSL paradigm is challenging, particularly comparing with existing $FedU^2$ [17], *FedX* [5] with identical architectures.

Although both *Hetero-SSFL* [20] and *FedFoA* [19] consider the heterogeneous client models, they can not learn a global representation model. In order to aggregate the knowledge from the clients to form global class representations, some tricky challenges arise. (1) *Deviated representation abilities.* Even for the same data samples, the different models might encode them into different latent spaces with *deviated representation abilities*. For example, *client A* and *client B* all have images with *dog, cat, tiger*, but *client model A* can encode *cat, tiger* well into different clusters, *client model B* can only learn better representations of *dog*. So *how could global representation models take advantage of the best of both client models*? (2) *Inconsistent representation spaces.* The skewed class distributions across clients lead to inconsistent representation spaces. For example in Fig. 1, comparing with *client A*, *client C* has different kinds of images with *dog, cat, airplane*. Thus *how to make global representation models encode the whole classes from all the clients well in a unified space*? Therefore, different from the existing works, our goal is to break the gaps caused by the hybrid heterogeneity, which can learn the high-quality global representation model in federated self-supervised learning.

Along this line, we propose *FedMKD*, a multi-teacher knowledge distillation based resource-adaptive Fed-SSL framework, which can learn global representations over all classes from heterogeneous clients. First, an adaptive knowledge integration module is introduced to learn high-quality representations from all the heterogeneous models with deviated representation abilities. Then in order to encode all classes from clients in a unified space, the global model uses the weighted combination of self-supervised loss and distillation loss to update. Besides, the global knowledge anchored alignment module is applied within the server to eliminate the inconsistency in representation spaces and reduce the burden on the clients. It uses global knowledge to additionally update the local models, which can not only make the local representation spaces close to the global space but also improve the representation capability of both local models and the global ones. Code is available at https://github.com/limee-sdu/FedMKD. The main contributions of this paper can be summarized as follows.

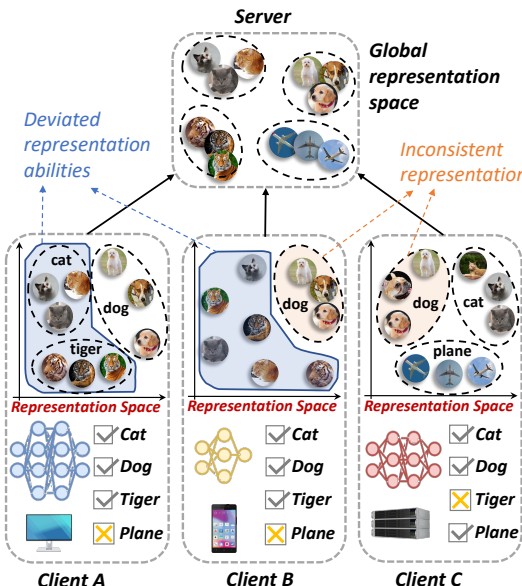

Figure 1: Illustrations of main challenges in resource-aware Fed-SSL.

- In resource-aware Fed-SSL, we are the first to delve into global class representation learning through revealing the deviated representation abilities and inconsistent representation spaces caused by the heterogeneous architectures and class skew.

- We design a multi-teacher knowledge distillation framework, namely *FedMKD*, to adaptively aggregate positive knowledge from heterogeneous models with deviated representation abilities. Through combining the self-supervised loss and the distillation loss, *FedMKD* can encode skewed classes into a unified space.

- Extensive experiments conducted on *CIFAR-10* and *CIFAR-100* show the representation abilities over all classes of the *FedMKD* perform better than state-of-the-art baselines. Our algorithm can improve 4.22% and 5.31% separately on the two chosen datasets.

## 2 Related work

The federated self-supervised learning aims to learn high-quality representations from clients without large labeled datasets [11]. From the beginning, several works [12, 23] simply combine federated learning with self-supervised methods. Besides, *FedU* [33] designs a communication-efficient mech-

Table 1: Comparison of federated self-supervised learning methods.

| Method | Global model | Global model size | Model Heterogeneity | Deviated representation ability | Inconsistent representation space | Theoretical Analysis |
|---|---|---|---|---|---|---|
| FedU [33] | ✓ | = Client | ✗ | ✗ | ✗ | ✗ |
| FedEMA [34] | ✓ | = Client | ✗ | ✗ | ✗ | ✗ |
| L-DAWA [21] | ✓ | = Client | ✗ | ✗ | ✗ | ✗ |
| FedX [5] | ✓ | = Client | ✗ | ✗ | ✗ | ✗ |
| FedCA [30] | ✓ | = Client | ✗ | ✗ | ✓ | ✗ |
| FLPD [29] | ✓ | = Client | ✓ | ✗ | ✗ | ✗ |
| $FedU^2$ [17] | ✓ | = Client | ✗ | ✗ | ✓ | ✓ |
| FedFoA [19] | ✗ | - | ✓ | ✗ | ✓ | ✗ |
| Hetero-SSFL [20] | ✗ | - | ✓ | ✗ | ✓ | ✓ |
| **FedMKD(ours)** | ✓ | ≥ Client | ✓ | ✓ | ✓ | ✓ |

anism by only aggregating the online encoders under non-IID data. *FedUTN* [16] is proposed to use the aggregated online networks for the target network updating in the self-supervised framework. *L-DAWA* [21] proposes the layer-wise divergence aware weight aggregation to mitigate the influence of client bias. *FedEMA* [34] considers the divergence-aware moving average updating in clients, measuring the divergence between local models and global model. *FedX* [5] proposes a unsupervised federated learning framework to learn representations through a two-sided distillation method. However, all the above works intuitively gain the global model through parameters average due to the identical client models, which can not be applied in the heterogeneous clients setting directly. In addition, although *FedCA* [30] address the misaligned and inconsistent representation challenges by gathering features from clients, inducing potential privacy problems. *FLPD* [27, 29] introduces distillation method based similarity between prototypes from a labeled public dataset to update the local model. $FedU^2$ [17] focuses on mitigating representation collapse entanglement and obtaining unified representation spaces.

Considering heterogeneous client models in federated self-supervised learning, *Hetero-SSFL* [6] introduces linear-CKA to align lower-dimensional representations between the local model and global model without architectural constraints. *FedFoA* [19] designs a factorization-based method to extract the cross-feature relation matrix from the local representations for aggregation. However, both *Hetero-SSFL* and *FedFoA* can not learn a global representation model considering the hybrid heterogeneity, which is the main focus of our work. The comparison details are shown in Table 1.

## 3 Preliminaries

The goal of federated unsupervised learning is to learn the generalized representation for some downstream tasks from several distributed unlabeled data sources. A federated learning setting consists of a central server and $N$ clients. Each client $k$ contains a local unlabeled dataset $\mathcal{D}_k$, and the server contains a public unlabeled dataset $\mathcal{D}_{pub}$. The local objective at $k$-th client is

$$\min_{\theta_k} F(\theta_k) = \mathbb{E}_{\xi_k \sim \mathcal{D}_k}[\mathcal{L}_k(\theta_k, \xi_k)], \tag{1}$$

to minimize the expected local loss of client $k$ on local dataset $\mathcal{D}_k$ and $\xi_k$ is the unlabeled data. In traditional FL settings, it's assumed that $\{\theta_k\}$ are identical and gain the global model using $\theta = \sum_{k=1}^{N} p_k \theta_k$, where $p_k$ is the weight of $k$-th client. But in real-world cross-device scenarios, each client might have a unique model and the architecture of the model might be different, which means that traditional aggregation methods are not available. We assume that $\theta_k$ is not similar to others, and use $\{\theta_k\}$ to collaborate in training the larger global model $\theta$ in server. Here we define the global update function is $\theta_t = \mathcal{G}(\theta_{t-1}; \theta_1, \cdots, \theta_N)$. Our final aim is to optimize the global goal

$$\min_{\theta} F_{global}(\theta) = \mathbb{E}_{\xi \sim \mathcal{D}_{global}}[\mathcal{L}(\theta, \xi)], \tag{2}$$

where $\mathcal{L}(\theta, \xi)$ is global loss function in server and $\xi$ is the unlabeled data sampled from global dataset.

## 4 Designed FedMKD Method

We propose a multi-teacher knowledge distillation based federated self-supervised learning framework *FedMKD*, which is shown in Fig. 2. In *FedMKD*, besides the local self-supervised learning (Sec. 4.1),

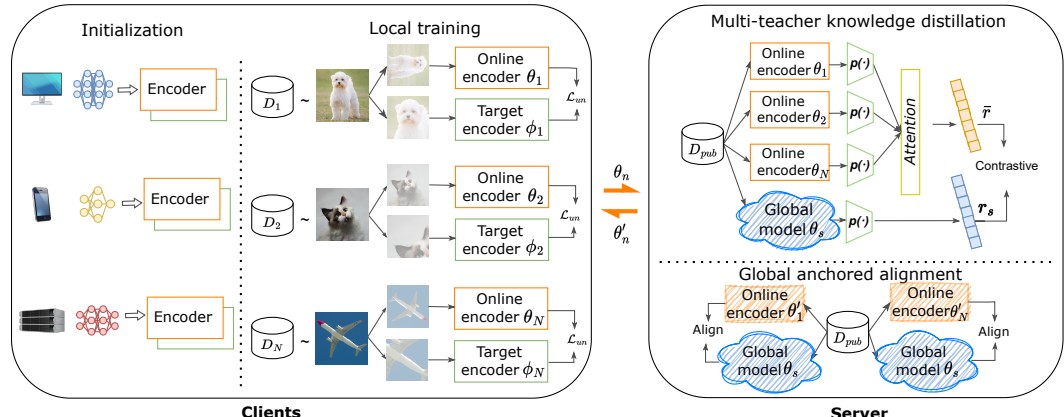

Figure 2: The overall framework of *FedMKD*. Clients initialize the model architecture based on the local resource, then self-supervised train the local model using unlabeled local data. The server uses the multi-teacher adaptive knowledge integration distillation to aggregate positive local knowledge to train the global model and then updates local models again according to the alignment module.

we design a multi-teacher adaptive knowledge integration distillation module to adaptive determine the weight of the representations from heterogeneous local models with deviated representation abilities. The distilled loss combined with the global self-supervised loss, we can gain the weighted combined loss to update the global model, so that the global model can encode all classes from clients in a unified space (Sec. 4.2). And the global knowledge anchored alignment could improve the representation capability of clients and further benefit the global model training (Sec. 4.3). In addition, we provide the theoretical analysis of our algorithm in Appendix B.

## 4.1 Self-supervised model training

Each client performs self-supervised contrastive learning using an asymmetric Siamese network, inspired by BYOL [3]. The model $\mathcal{M}$ comprises an online encoder $\theta$ and a target encoder $\phi$, both sharing the same architecture, with the online network incorporating an additional predictor $p$. That is $\mathcal{M} = \{p(\phi(\cdot)), \theta(\cdot)\}$. Given an unlabeled image $x$, we can obtain two augmented views, $v$ and $v'$, serving as inputs to online and target networks, respectively. The loss function is defined as follows:

$$\mathcal{L}_{self} = \left\| \frac{p(r)}{\|p(r)\|} - \frac{r'}{\|r'\|} \right\|^2, \tag{3}$$

where $r = \theta(v)$ and $r' = \phi(v')$. This loss encourages the online network to produce representation $p(r)$ that is similar to the positive sample generated by the target network $r'$. We then exchange the views, feeding $v'$ to the online network and $v$ to the target network, to compute $\mathcal{L}'_{self}$. At each training step, we use stochastic gradient descent to minimize $\tilde{\mathcal{L}}_{self} = \mathcal{L}_{self} + \mathcal{L}'_{self}$ to update the online network $\phi$ alone,

$$\theta \leftarrow \theta - \eta \nabla_\theta \tilde{\mathcal{L}}_{self}. \tag{4}$$

The target network helps to provide regression targets to train the online network. Choosing $\alpha \in [0, 1]$ as the target decay rate, we employ the exponential moving average (EMA) of the online network to update $\phi$:

$$\phi \leftarrow \alpha\phi + (1 - \alpha)\theta. \tag{5}$$

Using this self-supervised training method, the model learns intricate representations from unlabeled data, capturing high-level features and patterns inherent in the dataset.

## 4.2 Multi-teacher adaptive knowledge integration distillation.

In contrast to homogeneous federated learning, the presence of model heterogeneity poses a challenge: direct aggregation of local models into a global model is not feasible. To overcome this, we design a multi-teacher knowledge distillation mechanism to transfer local knowledge to the server.

Given a batch of data $\mathcal{B}$, the representation from the teacher model is denoted as $r_t$ and that from the student model as $r_s$, the knowledge distillation loss is defined as follows:

$$L_{distill} = -log \frac{exp(sim(r_{s,i}, r_{t,i})/\tau)}{exp(sim(r_{s,i}, r_{t,i})/\tau) + \sum\limits_{k \in \{\mathcal{B}-i\}} exp(sim(r_{s,i}, r_{s,k})/\tau)}, \tag{6}$$

where $\tau$ is the temperature parameter controlling entropy and $sim(\cdot)$ is the similarity function between two representations.

Then we extend this knowledge distillation learning method to multi-teacher. Although the data is heterogeneous, the knowledge of each local model is valuable, each local model captures the unique characteristics of local data. Our goal is to integrate the positive knowledge of all clients to guide the global model in learning a general representation of unlabeled data. We design a multi-teacher adaptive knowledge integration distillation that can adaptively weigh the representations from clients.

Given a sample $x_i$, representation from the $n$-th local model is $R_{n,i} \in \mathbb{R}^d$ where $d$ is the dimension of the representation. Following [3], a fully connected layer is employed to project the representation into a lower-dimensional space, enhancing the discriminate power of the learned representations. So, we map the representation from the global model $R_{s,i}$ into the same lower latent space, obtaining

$$r_{s,i} = g(R_{s,i}), r_{n,i} = g(R_{n,i}), \tag{7}$$

where $r_{n,i}, r_{s,i} \in \mathbb{R}^k$, $k$ is the dimension of the new latent space and $g(\cdot)$ is the projector.

In addition, we introduce an adapter module to learn instance-level teacher importance weights for knowledge integration. After getting $r_{n,i}$, an attention block is used to generate the weighted sum of them. In this context, representation from global model $r_{s,i}$ is treated as the *query*, while those from local models $\tilde{R} = [r_{1,i}, r_{2,i}, \ldots, r_{N,i}]^T$ is treated as the *key* and *value*. Treating representations from the global model as *query* ensures consistency in knowledge transfer. The attention mechanism computes attention scores to understand the relevance of each local model's representation to the global model's query. The aggregated representation is:

$$\bar{r}_i = Attn(r_{s,i}, \tilde{R}) = softmax(\frac{r_{s,i} \cdot \tilde{R}}{\sqrt{k}})\tilde{R}, \tag{8}$$

where $\bar{r}_i$ means the aggregated representation, and $Attn(\cdot)$ denotes the attention block.

Returning to the knowledge distillation for unlabeled data proposed earlier, we treat the aggregate representation as the positive sample, and the remaining samples in the batch as the negative sample. The adaptive weight multi-teacher knowledge distillation loss function is expressed as follows:

$$L_{distill} = -log \frac{exp(sim(r_{s,i}, \bar{r}_i)/\tau)}{exp(sim(r_{s,i}, \bar{r}_i)/\tau) + \sum\limits_{j \neq i} exp(sim(r_{s,i}, r_{s,j})/\tau)}. \tag{9}$$

Above all, the weighted combined loss for the global model is presented as:

$$L_{server} = L_{self} + \gamma L_{distill}, \tag{10}$$

where $\gamma$ is a hyper-parameter controlling the weight of the distillation process.

### 4.3 Global knowledge anchored alignment

As we mentioned, the representations from different models are inconsistent and the representation abilities of models are also deviated. So we introduce the global knowledge anchored alignment mechanism that each local model uses the global model as an anchor. It ensures that the local representation spaces are closer to the global ones.

Unlike methods such as FedX [5] and MOON [15], which align local models to the global model locally, our approach aims to train a better global encoder tailored for resource-constrained federated learning scenarios. Those methods are not available when clients cannot afford to store or infer the global model locally. So we transfer this alignment process to the server.

After finishing the global model training, we construct local twin models in the server to realize the alignment under global view. Here, we use the global online network and local online network to

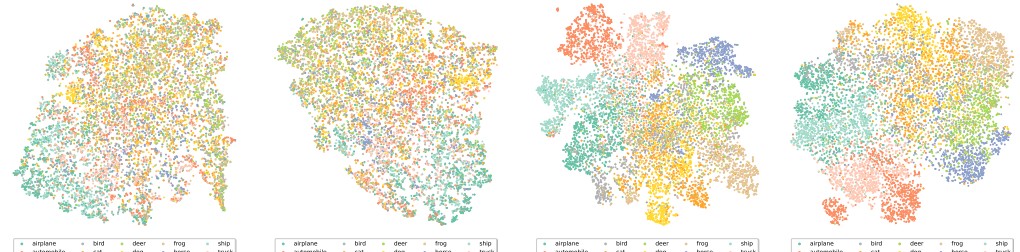

(a) Standalone training ResNet18 on Partial public dataset. (b) MOON on Partial public lic datset. (c) FedMKD on IID public dataset (d) FedMKD on Partial public dataset

Figure 3: T-SNE visualizations of hidden vectors from different models on CIFAR-10, the data distribution of clients is IID.

construct a new asymmetric siamese network called the twin of the original local model. The local online network $\theta_n$ is the online model, and the global online network $\theta_s$ is the target model, that is,

$$\{\theta'_n, \phi'_n\} \leftarrow \{\theta_n, \theta_s\} \tag{11}$$

and $\tilde{M} = \{\theta'_n, \phi'_n\}$. The training loss is updated to

$$\mathcal{L}_{align} = -log \frac{exp(sim(\theta'_n(i), \phi'_n(i))/\tau)}{\sum_{i \in B} exp(sim(\theta'_n(i), \phi'_n(i))/\tau)}. \tag{12}$$

According to the idea of contrastive learning, the representations learned by the online network become more consistent with the knowledge captured by the target network. This global knowledge anchored contrastive learning suggests that the global model's knowledge is used as a positive example for the local model to train itself, thus making it more consistent with the target global network. Aligning local models with the global view representation helps create a comprehensive understanding of the overall data distribution. The process refines the knowledge acquired locally, ensuring that it contributes meaningfully to the overall federated learning process. Then, we use stochastic gradient descent to minimize $\mathcal{L}_{align}$ to update the $\theta'_n$,

$$\theta'_n \leftarrow \theta'_n - \eta \nabla \mathcal{L}_{align}. \tag{13}$$

Once the global model anchored alignment is finished, the server will send the online network of the local twin network $\theta'$ to the corresponding client to update the local model. The local online network $\theta_n$ is replaced by the server-updated network $\theta'_n$. The target network is not replaced to retain more local knowledge and stabilize model training:

$$\{\theta_{q,n}, \phi_{q,n}\} \leftarrow \{\theta'_{q-1,n}, \phi_{q-1,n}\} \tag{14}$$

so that, the local model can benefit from the alignment process and align to the representation under global view, which can further use the local data to train the model.

## 5 Experiments

In this section, we evaluate the representations learned from our proposed global model *FedMKD* on *CIFAR-10* and *CIFAR-100*. We first describe the experimental setup and baselines, and then analyze the performance in comparison to other methods. Due to the space limitation, further hyperparameter analysis and communication cost analysis are represented in Appendix D.

### 5.1 Experimental setup

We use *CIFAR-10* and *CIFAR-100* [13] datasets to train all the models. Both of them contain 50,000 training images and 10,000 testing images. To construct the public dataset, we sample 4000 data samples from the training set, then divide the remaining data into $N$ partitions to simulate $N$ clients.

Table 2: Top-1 accuracy comparison under linear probing on CIFAR datasets with best model performance in bold and second-best results with underlines. '-' means this method is not suitable for the experiment setting.

| Method | Pub. | CIFAR-10 (%) | | | CIFAR-100 (%) | | |
|---|---|---|---|---|---|---|---|
| | | Class | Dir($\beta$=0.5) | IID | Class | Dir($\beta$=0.5) | IID |
| Std. ResNet18 | | | $51.09 \pm 0.04$ | | | $25.35 \pm 0.02$ | |
| FedMD | | $45.28 \pm 0.02$ | $45.93 \pm 0.02$ | $46.21 \pm 0.02$ | $23.25 \pm 0.05$ | $22.46 \pm 0.04$ | $23.20 \pm 0.03$ |
| FedDF | | $46.94 \pm 0.04$ | $48.04 \pm 0.02$ | $48.74 \pm 0.08$ | $23.07 \pm 0.03$ | $22.73 \pm 0.03$ | $21.57 \pm 0.01$ |
| MOON-KL | | $44.93 \pm 0.05$ | $45.84 \pm 0.03$ | $46.51 \pm 0.03$ | $21.26 \pm 0.03$ | $21.34 \pm 0.01$ | $21.82 \pm 0.02$ |
| MOON | IID | $53.35 \pm 0.03$ | $53.71 \pm 0.04$ | $55.14 \pm 0.02$ | $27.82 \pm 0.01$ | $26.84 \pm 0.03$ | $26.70 \pm 0.03$ |
| FedET | | $56.42 \pm 0.02$ | $59.38 \pm 0.03$ | $61.43 \pm 0.02$ | $29.11 \pm 0.03$ | $26.98 \pm 0.01$ | $24.48 \pm 0.02$ |
| FedU/FedEMA | | - | - | - | - | - | - |
| Hetero-SSFL | | $\underline{59.13 \pm 0.02}$ | $\underline{64.04 \pm 0.04}$ | $\underline{65.61 \pm 0.07}$ | $\underline{30.84 \pm 0.10}$ | $\underline{29.63 \pm 0.06}$ | $\underline{28.89 \pm 0.06}$ |
| FedMKD | | $\mathbf{64.81 \pm 0.02}$ | $\mathbf{66.98 \pm 0.06}$ | $\mathbf{69.07 \pm 0.04}$ | $\mathbf{36.33 \pm 0.01}$ | $\mathbf{35.59 \pm 0.07}$ | $\mathbf{35.94 \pm 0.02}$ |
| Std. ResNet18 | | | $50.15 \pm 0.02$ | | | $24.97 \pm 0.01$ | |
| FedMD | | $47.16 \pm 0.03$ | $46.39 \pm 0.04$ | $45.93 \pm 0.03$ | $23.95 \pm 0.05$ | $23.14 \pm 0.03$ | $22.47 \pm 0.01$ |
| FedDF | | $52.59 \pm 0.05$ | $53.50 \pm 0.03$ | $54.17 \pm 0.05$ | $27.21 \pm 0.02$ | $27.31 \pm 0.04$ | $27.05 \pm 0.04$ |
| MOON-KL | | $46.41 \pm 0.03$ | $46.81 \pm 0.03$ | $45.89 \pm 0.01$ | $21.73 \pm 0.01$ | $20.97 \pm 0.03$ | $22.27 \pm 0.04$ |
| MOON | Par. | $54.31 \pm 0.04$ | $54.54 \pm 0.02$ | $52.94 \pm 0.04$ | $27.00 \pm 0.04$ | $27.27 \pm 0.01$ | $28.26 \pm 0.02$ |
| FedET | | $57.75 \pm 0.01$ | $57.08 \pm 0.01$ | $58.59 \pm 0.01$ | $29.38 \pm 0.01$ | $28.12 \pm 0.02$ | $\underline{29.61 \pm 0.01}$ |
| FedU/FedEMA | | - | - | - | - | - | - |
| Hetero-SSFL | | $\underline{63.20 \pm 0.08}$ | $\underline{61.93 \pm 0.04}$ | $\underline{61.15 \pm 0.07}$ | $\underline{30.94 \pm 0.05}$ | $\underline{29.92 \pm 0.03}$ | $29.56 \pm 0.03$ |
| FedMKD | | $\mathbf{66.39 \pm 0.09}$ | $\mathbf{67.60 \pm 0.04}$ | $\mathbf{65.88 \pm 0.03}$ | $\mathbf{35.82 \pm 0.02}$ | $\mathbf{35.55 \pm 0.05}$ | $\mathbf{34.38 \pm 0.02}$ |

To assess the validity of the public dataset, we use two sampling methods to construct it. First, we use a random sampling method over all classes to generate public dataset 'IID'. And for the public dataset 'Partial', data is selected randomly from 40% classes in two datasets.

We utilize three settings to simulate heterogeneous data distributions among all the clients. For the IID setting, each client contains the same number of samples from all classes. For the class setting, each client only has $10/N$ and $100/N$ classes on two datasets and the classes between clients have no overlap. For the non-IID setting, data heterogeneity levels are described by the Dirichlet distribution $Dir(\beta)$ [10], where smaller $\beta$ represents stronger heterogeneity levels, here we choose $\beta = 0.5$.

Regarding the self-supervised learning framework design within each client, we use ResNet18 [9] and VGG9 [22] as the encoder network and Multi-Layer Perception (MLP) as the predictor. In order to construct the model heterogeneous setting, 2 clients train the Resnet18 encoder while 3 clients use the VGG9. And for the global representation model, Resnet18 is selected as the encoder in server.

## 5.2 Baselines and evaluation methods

Firstly, we select several federated knowledge distillation frameworks *FedMD* [14], *FedDF* [18], *FedET* [2], *MOON* [15], *MOON-KL* that use unlabeled public dataset for distillation. We then replaced the local model with a self-supervised model to evaluate the process of knowledge distillation in our method. And several federated self-supervised learning frameworks *FedU* [33], *FedEMA* [34], *Hetero-SSFL* [20] are also chosen as baselines. To verify the client's knowledge can improve the global model, we also train the global model separately on the public dataset, denoted as *Std. ResNet18*. Following *FedEMA* [30], we evaluate the performance of learned representations using linear and semi-supervised evaluation. Due to limited space, please refer to Appendix C for more details.

## 5.3 Performance Evaluation

Table 2 and 3 shows the linear evaluation results and semi-supervised evaluation results of *FedMKD* compared with all the baselines on *CIFAR-10* and *CIFAR-100*. We can gain the following observation.

On the whole, our *FedMKD* outperforms all baselines under different public dataset settings and different data heterogeneity level settings on both two datasets. Compared to the second-best results, *FedMKD* achieves significant improvement. On average, our model improves *CIFAR-10* by 4.22% and *CIFAR-100* by 5.31% under linear evaluation and gain 3.66% and 2.07% improvement on two dataset under semi-supervised evaluation.

The effectiveness of our multi-teacher adaptive knowledge integration distillation can be approved when compared with *FedMD*, *FedDF*, *MOON-KL* and *MOON*. Although these methods all designed

Table 3: Top-1 accuracy comparison on 1% of labeled data for semi-supervised learning on CIFAR datasets with best model performance in bold and second-best results with underlines. '-' means this method doesn't apply for the experiment setting.

| Method | Pub. | CIFAR-10 (%) | | | CIFAR-100 (%) | | |
|---|---|---|---|---|---|---|---|
| | | Class | Dir($\beta$=0.5) | IID | Class | Dir($\beta$=0.5) | IID |
| Std. ResNet18 | | | $46.84 \pm 0.25$ | | | $15.19 \pm 0.20$ | |
| FedMD | | $43.32 \pm 0.22$ | $44.20 \pm 0.18$ | $44.66 \pm 0.20$ | $15.88 \pm 0.17$ | $14.94 \pm 0.19$ | $15.34 \pm 0.12$ |
| FedDF | | $43.60 \pm 0.44$ | $44.13 \pm 0.16$ | $44.80 \pm 0.40$ | $14.39 \pm 0.20$ | $13.06 \pm 0.14$ | $12.90 \pm 0.07$ |
| MOON-KL | | $45.42 \pm 0.26$ | $46.61 \pm 0.21$ | $46.72 \pm 0.15$ | $16.25 \pm 0.06$ | $17.22 \pm 0.25$ | $16.07 \pm 0.04$ |
| MOON | IID | $49.96 \pm 0.24$ | $50.21 \pm 0.10$ | $51.78 \pm 0.28$ | $19.23 \pm 0.12$ | $17.21 \pm 0.13$ | $17.07 \pm 0.18$ |
| FedET | | $52.37 \pm 0.24$ | $56.57 \pm 0.17$ | $57.44 \pm 0.13$ | $19.70 \pm 0.08$ | $16.82 \pm 0.20$ | $15.68 \pm 0.18$ |
| FedU/FedEMA | | - | - | - | - | - | - |
| Hetero-SSFL | | $54.30 \pm 0.15$ | $58.73 \pm 0.54$ | $60.50 \pm 0.12$ | $20.04 \pm 0.40$ | $19.19 \pm 0.17$ | $18.82 \pm 0.17$ |
| FedMKD | | $\mathbf{59.65 \pm 0.28}$ | $\mathbf{61.78 \pm 0.40}$ | $\mathbf{64.06 \pm 0.32}$ | $\mathbf{22.57 \pm 0.12}$ | $\mathbf{22.13 \pm 0.11}$ | $\mathbf{22.07 \pm 0.15}$ |
| Std. ResNet18 | | | $46.42 \pm 0.12$ | | | $14.12 \pm 0.09$ | |
| FedMD | | $44.54 \pm 0.26$ | $43.61 \pm 0.13$ | $42.52 \pm 0.19$ | $17.32 \pm 0.17$ | $16.47 \pm 0.14$ | $16.31 \pm 0.07$ |
| FedDF | | $48.14 \pm 0.27$ | $48.74 \pm 0.12$ | $48.56 \pm 0.18$ | $17.01 \pm 0.04$ | $17.14 \pm 0.01$ | $16.95 \pm 0.01$ |
| MOON-KL | | $46.76 \pm 0.05$ | $46.92 \pm 0.05$ | $46.49 \pm 0.22$ | $16.21 \pm 0.27$ | $16.10 \pm 0.12$ | $16.94 \pm 0.12$ |
| MOON | Par. | $50.43 \pm 0.18$ | $51.99 \pm 0.34$ | $49.86 \pm 0.19$ | $18.64 \pm 0.21$ | $18.92 \pm 0.25$ | $19.29 \pm 0.08$ |
| FedET | | $52.75 \pm 0.07$ | $52.61 \pm 0.03$ | $54.64 \pm 0.14$ | $18.49 \pm 0.12$ | $18.16 \pm 0.13$ | $18.01 \pm 0.22$ |
| FedU/FedEMA | | - | - | - | - | - | - |
| Hetero-SSFL | | $59.95 \pm 0.34$ | $58.31 \pm 0.50$ | $58.35 \pm 0.20$ | $20.72 \pm 0.14$ | $20.30 \pm 0.32$ | $19.62 \pm 0.08$ |
| FedMKD | | $\mathbf{61.55 \pm 0.19}$ | $\mathbf{63.10 \pm 0.28}$ | $\mathbf{61.08 \pm 0.19}$ | $\mathbf{22.44 \pm 0.19}$ | $\mathbf{22.21 \pm 0.17}$ | $\mathbf{20.94 \pm 0.25}$ |

new federated knowledge distillation frameworks based on unlabeled public dataset, since the original local model is supervised, they prefer using the class information from logits to distill. When observing the result of *FedET*, we find that although it also designs a larger global model in server which improves the model a lot, the final result is not satisfied. This is also because it designs a distillation method based on knowledge of probability distribution over classes. Next, compared with the federated self-supervised method, our *FedMKD* also achieves better performance. *FedU* and *FedEMA* are not applicable in model heterogeneity setting, so we cannot evaluate their effectiveness. *Hetero-SSFL* gain the best performance among all baselines but is worse than ours. That's because it aims to train personalised client models. Only the alignment module cannot hold the inconsistent representation space perfectly. But our global model can directly generate representation from the global model, it avoids using the representation from inconsistent clients.

Apart from the client data heterogeneity, we consider the influence of the public dataset distribution. Here, we construct two public datasets, one is 'IID' to the whole data distribution and the other only has partial classes. In both two settings, our *FedMKD* also gets the best performance. The overall performance of 'IID' public dataset is better than 'Partial' setting. That's because the global model adapts the self-supervised learning on public dataset, and the diversity of the sample is important, which can help the model explore a wide range of features and patterns present in the data. And it's observed that when the public dataset is 'IID', the performance increases with the decrease of the data heterogeneity level for *CIFAR-10*, but it doesn't apply to the *CIFAR-100*. It's because there are too many classes in *CIFAR-100* and the number of samples in each class is not efficient.

In order to evaluate the effectiveness of *FedMKD*, we use the dimensionality reduction algorithm t-sne to visualize the representation on the test dataset of *CIFAR-10* from different encoders. As shown in Fig. 3 (b)(c)(d), the global models in *FedMKD* trained on both 'IID' and 'Partial' public datasets both achieve better clustering results than Standalone training and *MOON*. These results further verify our model can gain better generalized representations although the representation spaces of clients are inconsistent. There's also an averaged global model in *MOON*, but it cannot tackle the problem of inconsistent spaces well using the average method, so it only gains a poor clustering performance. Additionally, the performance on 'IID' public dataset is better than 'paritial' ones from the observation of cluster performance. This suggests that the number of classes seen by the global model also affects how well the global model can encode all classes in a unified space. Comparing the class distributions in Fig. 3 (c) and (d), we can find that although these two global models are trained on different public datasets, the final cluster layout is similar, which can further prove that our global model can encode all classes from clients even if it never sees some classes during the training.

Table 4: Experimental results on ablation studies of FedMKD with best model performance in bold.

| Method | CIFAR-10 (%) | | CIFAR-100 (%) | |
|---|---|---|---|---|
| | **Class** | **IID** | **Class** | **IID** |
| Standalone training | $51.09 \pm 0.04$ | | $25.35 \pm 0.02$ | |
| FedMKD$_{KL}$ | $43.88 \pm 0.19$ | $46.24 \pm 0.08$ | $19.56 \pm 0.18$ | $15.45 \pm 0.05$ |
| FedMKD$_{w/o\ adaptive}$ | $46.14 \pm 0.05$ | $47.29 \pm 0.08$ | $21.77 \pm 0.02$ | $22.76 \pm 0.03$ |
| FedMKD$_{w/o\ alignment}$ | $61.85 \pm 0.04$ | $62.34 \pm 0.09$ | $34.97 \pm 0.04$ | $29.91 \pm 0.02$ |
| FedMKD | $\mathbf{64.81 \pm 0.02}$ | $\mathbf{69.07 \pm 0.04}$ | $\mathbf{36.33 \pm 0.01}$ | $\mathbf{35.94 \pm 0.02}$ |

## 5.4 Improvement of clients

In *FedMKD*, the global knowledge anchored alignment module is used to align the client model in the server which can further transfer the global knowledge to the client and incentivize the clients to participate in the federated learning. To evaluate the improvement of the clients, the local models which are standalone training locally are compared with our local models. As shown in Fig. 4, the client performance in our *FedMKD* framework is better than local standalone training, regardless of the archi-

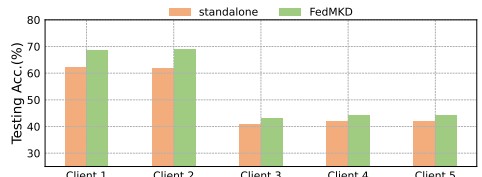

Figure 4: Improvement of clients after involving our proposed *FedMKD*.

tecture of the local model. Especially for clients with ResNet18, it improves 6.73% on average. It's concluded that clients benefit from federated training by contributing to global training.

## 5.5 Validation of inconsistent representation spaces

As we mentioned, the representation spaces between different clients are inconsistent because of data heterogeneity. To validate this opinion, we use Linear Discriminant Analysis (LDA) to reduce dimensionality in order to visualize the distribution of the representations. Here the data distribution of clients is Class. In Fig. 5 left, '○' and '×' denote representations of Client A and Client B, respectively. And different colors denote different classes. We can observe that in Fig. 5 left the classes 'cat' and 'dog' almost overlap while they are from different clients, which verifies that the inconsistent representation spaces did exactly exist. And Fig. 5 right shows the visualization result of the represen-

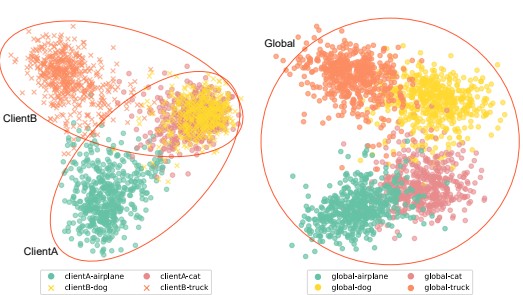

Figure 5: LDA visualizations of hidden vectors from different models on CIFAR-10.

tation from the global model. It's clear that the classes 'cat' and 'dog' are embedded in different positions in global space, which proves that although the local representations are inconsistent, our global model can learn a good representation.

## 5.6 Ablation experiment

In order to investigate the effectiveness of different parts of *FedMKD*, we design these comparison experiments:

- *Standalone training*: The global model is trained alone using the public dataset, without the knowledge from client models.

- *FedMKD$_{KL}$*: The global distillation function is replaced by the KL-divergence function to measure the similarity of the aggregated representation $\bar{r}$ and global representation $r_s$.

- *FedMKD$_{w/o\ adaptive}$*: The adaptive knowledge integration module is removed, each local representation has the same weight to generate the aggregated representation.

Table 5: Experimental results on scalability studies of FedMKD.

| Method | Client number | | |
|---|---|---|---|
| | 5 | 10 | 30 |
| Linear result | $67.79 \pm 0.04$ | $57.39 \pm 0.60$ | $53.85 \pm 0.03$ |
| Semi result | $63.37 \pm 0.51$ | $53.30 \pm 0.31$ | $48.97 \pm 0.86$ |

- $FedMKD_{w/o\ alignment}$: The global knowledge anchored alignment module is removed from *FedMKD*.

Above experiment is conducted on the both *CIFAR-10* and *CIFAR-100* dataset, using the IID public dataset. The results of the ablation study experiment are shown in Table 4. When comparing *FedMKD* with $FedMKD_{KL}$, significant performance drop can be observed when the knowledge distillation function is replaced by the KL-divergence function. Thus we can conclude that for self-supervised learning, we prefer performing knowledge distillation based on representation, but the KL-divergence cannot capture the distribution characteristics from them. Therefore, the appropriate distillation method is critically important in self-supervised learning due to the lack of labels. And a worse performance on both two datasets can be observed when we use the equal weight instead of the adaptive weight. Because the client models are heterogeneous, their representation capabilities are different and the representation spaces are also inconsistent, so intuitively average representations may reduce the information contained in the representation. Finally, when we remove the alignment module, the performance under each setting all decreases, which demonstrates that the alignment is not only beneficial to the local models, but also improves the whole training process.

### 5.7 Scalability of algorithm

In order to explore the scalability of our proposed algorithm *FedMKD*, we add the experiment that the number of clients is 5, 10, 30 on *CIFAR-10*, and 40% clients use the VGG model and 60% use ResNet18. And we repartition the data for each client under $\text{Dir}(\beta = 0.5)$ and set the public dataset distribution as 'IID'. The results are shown in Table 5. We can find that as the number of clients increasing, the performance decreases. The reason is that the total number of data is fixed, if the number of clients increases, the number of data in each client will decrease, which further affect the performance of the local model.

## 6 Conclusion

In this work, we focused on how to solve the deviated representation abilities and inconsistent representation spaces caused by the heterogeneous architectures and class skew in federated self-supervised learning. We proposed a multi-teacher knowledge based federated self-supervised learning framework *FedMKD* to learn a global model. Firstly, the adaptive knowledge integration module could learn high-quality representation knowledge from heterogeneous models. And the combination of the self-supervised loss and the distillation loss enabled the global model to encode all classes from clients in a unified space. Then a global knowledge anchored alignment module improved the local representation models in server and fed it back to corresponding clients. The experiments conducted on two datasets demonstrated that our proposed *FedMKD* was state-of-the-art and outperformed existing methods.

## Acknowledgement

This work was supported in part by the National Natural Science Foundation of China under Grant 62202273, 62176014, 92370204, in part by National Science Fund for Excellent Young Scholars of China under Grant 62122042, in part by Shandong Provincial Natural Science Foundation of China under Grant ZR2021QF044, in part by the Fundamental Research Funds for the Central Universities, in part by the Major Basic Research Program of Shandong Provincial Natural Science Foundation under Grant ZR2022ZD02, in part by the Joint Key Funds of National Natural Science Foundation of China under Grant U23A20302, in part by Guangzhou-HKUST(GZ) Joint Funding Program under Grant 2023A03J0008, in part by the Education Bureau of Guangzhou Municipality.

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

## Appendix

We provide more details about our work and experiments in the appendices:

- Appendix A: the additional details of the proposed algorithm FedMKD.
- Appendix B: the detailed proof of the convergence analysis of our proposed algorithm FedMKD.
- Appendix C: the details of experimental settings including datasets and federated simulations, reproduction details, baselines, evaluation methods and ablation experiment setting.
- Appendix D: additional experimental results including Appendix D.1, the overall performance under two evaluation methods; Appendix D.2, the communication and storage efficiency analysis; Appendix D.3 discusses the impact of hyperparameters.
- Appendix E: the details of limitations and broader impacts of this work.

## A    Algorithm description

We outline the `FedMKD` algorithm in Algorithm 1. In round $q$, the clients and server perform the following updates:

- Starting from the resource-adaptive model $\theta_{q,n,0}$, we update the local parameters for $t \in [T]$

$$\theta_{q,n,t} \leftarrow \theta_{q,n,t-1} - \eta \nabla l_{q,n,t}.$$
$$\phi_{q,n,t} \leftarrow \alpha \phi_{q,n,t-1} + (1-\alpha)\theta_{q,n,t}.$$

- After $T$ times local updates, upload the local parameter $\theta_{q,n,T}$.
- Server uses all local model to process multi-teacer adaptive knowledge integration distillation, update the global model for $t \in [T']$,

$$\theta_{q,s} \leftarrow \theta_{q,s} - \eta \nabla l_{q,s}. \tag{15}$$
$$\phi_{q,s} \leftarrow \alpha \phi_{q-1,s} + (1-\alpha)\theta_{q,s}. \tag{16}$$

- Server aligns all client models to the global model, for $n \in N$,

$$\theta'_{q,n} \leftarrow \theta'_{q,n} - \eta \nabla l'_{q,n}. \tag{17}$$

## B    Convergence Analysis

In this section, we show the convergence analysis of our *FedMKD*. Firstly, we give some commonly used assumptions in federated learning:

**Assumption 1** *(Lipschitz Condition). Every function $F(\cdot)$ is with L-Lipschitz gradient: $\forall n \in [N], \theta, \varphi \in R^d$*

$$\|\nabla F(\theta) - \nabla F(\varphi)\| \le L\|\theta - \varphi\| \tag{18}$$

**Assumption 2** *(Bounded variance). The stochastic gradients $\nabla F_n(\theta_{q,n,t}; \xi_{n,t})$ is an unbiased estimator of the gradient, with the variance bounded by $\sigma > 0$:*

$$\mathbb{E}_{\xi_{n,t} \sim D_n} \|\nabla F_n(\theta_{q,n,t}; \xi_{n,t}) - \nabla F_n(\theta_{q,n,t})\|^2 \le \sigma^2, \quad \forall q, n, t \tag{19}$$

In order to analyze the convergence rate of our proposed *FedMKD*, we firstly state some preliminary lemmas as follows:

**Lemma 1** *(Jensen's inequality). For any convex function $h$ and any variable $x_1, \ldots, x_n$ we have*

$$h\left(\frac{1}{n}\sum_{i=1}^{n} x_i\right) \le \frac{1}{n}\sum_{i=1}^{n} h(x_i). \tag{20}$$

*Especially, when $h(x) = \|x\|^2$, we can get*

$$\left\|\frac{1}{n}\sum_{i=1}^{n} x_i\right\|^2 \le \frac{1}{n}\sum_{i=1}^{n} \|x_i\|^2. \tag{21}$$

**Algorithm 1:** Algorithm of FedMKD

---

**1** **Initialize:** total number of clients $N$; Number of rounds $T$; Local data in clients $\{D_1, \ldots, D_N\}$; Public data in server $D_{pub}$; Local online network $\{\theta_1, \ldots, \theta_N\}$, target network $\{\phi_1, \ldots, \phi_N\}$; Server online network $\theta_s$, target network $\phi_s$; Learning rate $\eta$; Adapter network $Attn(\cdot)$. **for** $q = 1$ *to* $Q$ **do**

**2**     **LocalUpdate: for** $n = 1$ *to* $N$ *(all clients in parallel)* **do**

**3**        **if** $q = 1$ **then**

**4**           Random initialize $\theta_{q,n,0}$.

**5**        **else**

**6**           $\{\theta_{q,n,0}, \phi_{q,n,0}\} \leftarrow \{\theta'_{q-1,n}, \phi_{q-1,n,T}\}$.

**7**        **end**

**8**        **for** *epoch* $t = 1$ *to* $T$ **do**

**9**           $l_{q,n,t} = \mathcal{L}_{self}(\theta_{q,n,t-1}; \phi_{q,n,t-1}; D_n)$.

**10**           $\theta_{q,n,t} \leftarrow \theta_{q,n,t-1} - \eta \nabla l_{q,n,t}$.

**11**           $\phi_{q,n,t} \leftarrow \alpha \phi_{q,n,t-1} + (1 - \alpha)\theta_{q,n,t}$.

**12**        **end**

**13**        Upload $\theta_{q,n,t}$.

**14**     **end**

**15**     **ServerExecution:**

**16**     *# Multi-teacher adaptive knowledge integration distillation.*

**17**     **for** *epoch* $t = 1$ *to* $T'$ **do**

**18**        **for** *batch* $b \in D_{pub}$ **do**

**19**           **for** $n = 1$ *to* $N$ **do**

**20**              $r_{q,n} = g(\theta_{q,n,t}(b))$.

**21**           **end**

**22**           $\tilde{R} = [r_{q,1}, \cdots, r_{q,N}]$

**23**           $r_s = g(\theta_{q,s}(b))$.

**24**           $\bar{r} = Attn(r_s, \tilde{R})$

**25**           $l_{q,s} = \mathcal{L}_{self}(\theta_{q,s}; \phi_{q,s}; b) + \gamma \mathcal{L}_{distill}(\theta_{q,s}; r_s; \bar{r})$.

**26**           $\theta_{q,s} \leftarrow \theta_{q,s} - \eta \nabla l_{q,s}$.

**27**           $\phi_{q,s} \leftarrow \alpha \phi_{q-1,s} + (1 - \alpha)\theta_{q,s}$.

**28**        **end**

**29**     **end**

**30**     *# Alignment client models in server.*

**31**     **for** $n = 1$ *to* $N$ **do**

**32**        $\{\theta'_{q,n}, \phi'_{q,n}\} \leftarrow \{\theta_{q,n,T}, \theta_{q,s}\}$

**33**        $l'_{q,n} = \mathcal{L}_{align}(\theta'_{q,n}; \phi'_{q,n}; D_{pub})$.

**34**        $\theta'_{q,n} \leftarrow \theta'_{q,n} - \eta \nabla l'_{q,n}$.

**35**     **end**

**36**     Send $\theta'_{q,n}$ to client $n$.

**37** **end**

---

**Lemma 2** *For random variable $x_1, \ldots, x_n$ we have*

$$\mathbb{E}[\|x_1 + \cdots + x_n\|^2] \leq n\mathbb{E}[\|x_1\|^2 + \cdots + \|x_n\|^2]. \tag{22}$$

**Lemma 3** *For independent random variables $x_1, \ldots, x_n$ whose mean is 0, we have*

$$\mathbb{E}[\|x_1 + \cdots + x_n\|^2] = \mathbb{E}[\|x_1\|^2 + \cdots + \|x_n\|^2]. \tag{23}$$

Based on the above assumptions, we present the theoretical results for the non-convex problem.

**Lemma 4** *(Deviation bound of the optimization function) In each communication round, the function value in server reduce after $T'$ epochs and is bounded as:*

$$\mathbb{E}[F(\theta_{q,T'})] \leq \mathbb{E}[F(\theta_{q,0})] - (\eta - \frac{L\eta^2}{2}) \sum_{t=1}^{T'} \|\nabla F(\theta_{q,t})\|^2 + \frac{LT'\eta^2}{2}\sigma^2 \tag{24}$$

Lemma 4 indicates the deviation bound of the optimization function of the server in each global round.

**Theorem 1** *(Non-convex divergence for FedMKD) Let Assumption 1 to 3 hold and $\Delta = F(\theta_0) - F(\theta_{T'})$, given any $\delta > 0$, suppose that the learning rates satisfy $0 \leq \eta \leq 2/L$, after*

$$Q = \frac{2\Delta}{T'\delta(2\eta - L\eta^2) - T'L\eta^2\sigma^2} \tag{25}$$

*communication round, we have*

$$\frac{1}{QT'}\sum_{q=0}^{Q-1}\sum_{t=0}^{T'-1}\mathbb{E}[\nabla\mathcal{L}_{q,t}] \leq \delta \tag{26}$$

*the convergence can be guaranteed.*

*Proof.* Let's start the proof from $L$-Lipschitz condition:

$$F(\theta_{q,t+1}) \overset{(a)}{\leq} F(\theta_{q,t}) + \langle\nabla F(\theta_{q,t}), \theta_{q,t+1} - \theta_{q,t}\rangle + \frac{L}{2}\|\theta_{q,t+1} - \theta_{q,t}\|^2$$

$$= F(\theta_{q,t}) + \langle\nabla F(\theta_{q,t}), -\eta\nabla F(\theta_{q,t}, \xi_{q,t})\rangle + \frac{L}{2}\| - \eta\nabla F(\theta_{q,t}, \xi_{q,t})\|^2$$

where (a) is from Assumption 1. Taking expectation of both sides, we obtain

$$\mathbb{E}[F(\theta_{q,t+1})] \leq \mathbb{E}[F(\theta_{q,t})] - \eta\mathbb{E}[\|\nabla F(\theta_{q,t})\|^2] + \frac{L\eta^2}{2}\mathbb{E}[\|F(\theta_{q,t}, \xi_{q,t})\|^2]$$

$$= \mathbb{E}[F(\theta_{q,t})] - \eta\mathbb{E}[\|\nabla F(\theta_{q,t})\|^2] + \frac{L\eta^2}{2}\mathbb{E}[\|F(\theta_{q,t})\|^2 + \sigma^2]$$

$$\overset{(b)}{\leq} \mathbb{E}[F(\theta_{q,t})] - (\eta - \frac{L\eta^2}{2})\mathbb{E}[\|\nabla F(\theta_{q,t})\|^2] + \frac{L\eta^2}{2}\sigma^2,$$

where (b) follows from Assumption 2. Let's set the learning step at the start of training to $T'$,

$$\mathbb{E}[F(\theta_{q,T'})] \leq \mathbb{E}[F(\theta_{q,0})] - (\eta - \frac{L\eta^2}{2})\sum_{t=1}^{T'}\|\nabla F(\theta_{q,t})\|^2 + \frac{LT'\eta^2}{2}\sigma^2 \tag{27}$$

*Proof of Theorem 1.* According to Lemma 4,

$$\frac{1}{QT'}(\eta - \frac{\eta^2 L}{2})\sum_{q=0}^{Q-1}\sum_{t=0}^{T'-1}\mathbb{E}[\nabla\mathcal{L}_{q,t}] \leq \frac{1}{QT'}\sum_{q=0}^{Q-1}\mathbb{E}[F(\theta_{q,T'})] - \frac{1}{QT'}\sum_{q=0}^{Q-1}\mathbb{E}[F(\theta_{q,0})] + \frac{LT'\eta^2}{2}\sigma^2$$

$$\leq \delta(\eta - \frac{\eta^2 L}{2})$$

Therefore,

$$\frac{\Delta}{Q} \leq \delta(\eta - \frac{\eta^2 L}{2}) - \frac{LT'\eta^2}{2}\sigma^2,$$

which is equal to

$$Q = \frac{2\Delta}{T'\delta(2\eta - L\eta^2) - T'(L\eta^2\sigma^2)}.$$

# C Experiment supplements

## C.1 Datasets and federated simulations

We use *CIFAR-10* and *CIFAR-100* [13] datasets to train all the models. Both of them contain 50,000 training images and 10,000 testing images. To construct the public dataset, we sample 4000 data

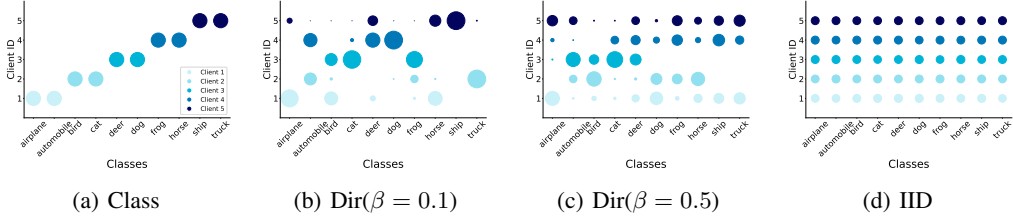

| (a) Class | (b) Dir($\beta = 0.1$) | (c) Dir($\beta = 0.5$) | (d) IID |

Figure 6: Illustrations of # of samples per class allocated to each client, for different distributions.

samples from the training set, then divide the remaining data into $N$ partitions to simulate $N$ clients. To assess the validity of the public dataset, we use two sampling methods to construct it. First, we use a random sampling method over all classes to generate public dataset 'IID'. And for the public dataset 'Partial', data is selected randomly from 40% classes in two datasets.

We utilise three settings to simulate heterogeneous data distributions among all the clients. For the IID setting, each client contains the same number of samples from all classes. For the class setting, each client only has $10/N$ and $100/N$ classes on two datasets and the classes between clients have no overlap. For the Non-IID setting, data heterogeneity levels are described by the Dirichlet distribution Dir($\beta$) [10], where smaller $\beta$ represents stronger heterogeneity levels. A value of $\beta = 0.1$ is chosen to simulate a high degree of heterogeneity, and $\beta = 0.5$ for a lower level. Fig. 6 shows the data distribution among clients on *CIFAR-10* dataset. The x-axis represents 10 classes and the y-axis is the total of 5 clients. The size of each circle denotes the number of samples for the specific class in the respective client.

## C.2   Reproduction details

Regarding the self-supervised learning framework design within each client, we use ResNet18 [9] and VGG9 [22] as the encoder network and Multi-Layer Perception (MLP) as the predictor. To compare with other baselines, we set the number of clients $N = 5$, and conduct the experiments for $R = 100$ rounds. In order to construct the model heterogeneous setting, 2 clients train the Resnet18 encoder while 3 clients use the VGG9. And for the global representation model, Resnet18 is selected as the encoder in server. The hyper-parameter $\gamma$ in the loss of global model is set to 0.9. During the training process, each client trains locally for $T = 5$ epochs while the server also distills for $T' = 5$ epochs. Finally, we set the target decay rate $\alpha = 0.99$, with a batch size of $B = 128$, and utilize SGD for optimization with a learning rate of $\eta = 0.032$. We implement all the methods in Python using EasyFL[32] based on PyTorch.

## C.3   Baselines

Firstly, we select several federated knowledge distillation frameworks that use unlabeled public dataset for distillation. We then replaced the local model with a self-supervised model to evaluate the process of knowledge distillation in our method.

- **FedMD**[14]: Each client trains convergence on the public dataset and then on local data. In each round, clients upload the embedding of the public dataset to the server. The server averages the embeddings and sends averaged embedding to clients. Clients first use average embedding to update local models on public dataset, then train on local data for a few epochs. There's no global model, the local models are evaluated to make comparison.

- **FedDF**[18]: Clients in FedDF train locally and upload the model parameter to server. The server uses all client models to compute the embedding of the public dataset and uses the average embedding to train the global model. But it has no global model, so we only test the performance of local models.

- **FedET**[2]: There is a global large model and several alternative small models on the server. Each client selects the appropriate small model for training locally and then uploads the parameter to the server. The server uses all the small models to compute representations of

the public dataset and updates the global model with the averaged representation. Then the server utilizes the representation from the global model to update all the small models.

- **MOON**[15]: Moon introduces a model-contrastive learning approach, treating the representation from the global model as positive knowledge to update the local models.
- **MOON-KL**: Instead of using the NT-Xent loss in MOON, MOON-KL utilises the KL-divergence function to measure the similarity between global representation and local representation. To make comparisons, both *MOON* and *MOON-KL* distribute the global model to each client, and then the client uses the local model to update the global model. The server aggregates the updated global models as the final global model.

And several federated self-supervised learning frameworks are also chosen as baselines.

- **FedU**[33]: Clients in FedU upload the local online encoder and predictor for server aggregation. Then they update the local online encoder using the aggregate ones, and dynamically update the predictor based on the divergence. But it cannot applied to the model heterogeneity setting.
- **FedEMA**[34]: FedEMA uses the aggregated local online network as a global model and measures the divergence between global and local networks. The divergence is then applied to the local exponential moving average update. It's also not applicable when models are heterogeneous.
- **Hetero-SSFL**[20]: Each client trains locally and uploads the kernel matrix over the public dataset. The server aggregates them and sends to clients. Then client uses linear-CKA normalization term to align to the averaged kernel matrix. There's no global model, so we just test local models' performance.

To verify that the client's knowledge can improve the global model, we also train the global model separately on the public dataset, denoted as *Std. ResNet18*.

## C.4 Evaluation methods

Following [33] and [30], we evaluate the performance of learned representations using linear and semi-supervised evaluation. For linear evaluation, all the models are trained without any supervised labels. Subsequently, the model encoder is then frozen and the representations are utilized for training a new classifier over 200 epochs. For semi-supervised evaluation, we consider the scenario that only a small subset of the data has a label, here only 1% data are labeled. Then, different from freezing the encoder, we fine-tune the whole model with a new classifier using the labeled data for 100 epochs. For *FedMD*, *FedDF* and *Hetero-SSFL*, only local models can be evaluated. So we use the averaged representation from local models to do the linear evaluation. But in semi-supervised evaluation, we need to fine-tune the whole model, so we evaluate each client's performance alone and use the average result as the final result.

# D Additional Experiments

## D.1 Numerical Results

Table 6, Table 7 and Table 8, Table 9 shows the linear evaluation results and semi-supervised evaluation results of *FedMKD* compared with all the baselines on *CIFAR-10* and *CIFAR-100*. Comparing to the result in main text, we extra add the result on $\text{Dir}(\beta = 0.1)$ to prove the efficiency of our algorithm totally.

## D.2 Communication and storage efficiency

Considering the experimental setting, the communication and storage capabilities are limited, so we analyze the communication and computation efficiency of *FedMKD* compared to *Hetero-SSFL* and all the experiments maintain the original settings. Firstly, we calculate that the memory of ResNet18 and VGG9 is 42.63MB and 16.38MB separately. We use $S(\theta)$ to denote the storage cost of model $\theta$.

About the communication cost, the client in *FedMKD* only needs to upload the online encoder to the server, and download the updated encoder, so the cost per communication is equal to the ResNet18 or

Table 6: Top-1 accuracy comparison under linear probing on CIFAR-10 datasets with best model performance in bold and second-best results with underlines. '-' means this method is not suitable for the experiment setting.

| Method | Public dataset | CIFAR-10 (%) | | | |
|---|---|---|---|---|---|
| | | Class | Dir($\beta$=0.1) | Dir($\beta$=0.5) | IID |
| Std. ResNet18 | | $51.09 \pm 0.04$ | | | |
| FedMD | | $45.28 \pm 0.02$ | $45.48 \pm 0.01$ | $45.93 \pm 0.02$ | $46.21 \pm 0.02$ |
| FedDF | | $46.94 \pm 0.04$ | $47.05 \pm 0.06$ | $48.04 \pm 0.02$ | $48.74 \pm 0.08$ |
| MOON-KL | | $44.93 \pm 0.05$ | $45.59 \pm 0.02$ | $45.84 \pm 0.03$ | $46.51 \pm 0.03$ |
| MOON | IID | $53.35 \pm 0.03$ | $53.52 \pm 0.03$ | $53.71 \pm 0.04$ | $55.14 \pm 0.02$ |
| FedET | | $56.42 \pm 0.02$ | $57.48 \pm 0.01$ | $59.38 \pm 0.03$ | $61.43 \pm 0.02$ |
| FedU/FedEMA | | - | - | - | - |
| Hetero-SSFL | | $\underline{59.13 \pm 0.02}$ | $\underline{62.62 \pm 0.14}$ | $\underline{64.04 \pm 0.04}$ | $\underline{65.61 \pm 0.07}$ |
| FedMKD | | $\mathbf{64.81 \pm 0.02}$ | $\mathbf{65.96 \pm 0.03}$ | $\mathbf{66.98 \pm 0.06}$ | $\mathbf{69.07 \pm 0.04}$ |
| Std. ResNet18 | | $50.15 \pm 0.02$ | | | |
| FedMD | | $47.16 \pm 0.03$ | $46.87 \pm 0.04$ | $46.39 \pm 0.02$ | $45.93 \pm 0.03$ |
| FedDF | | $52.59 \pm 0.05$ | $52.71 \pm 0.02$ | $53.50 \pm 0.03$ | $54.17 \pm 0.05$ |
| MOON-KL | | $46.41 \pm 0.03$ | $47.03 \pm 0.02$ | $46.81 \pm 0.03$ | $45.89 \pm 0.01$ |
| MOON | Partial | $54.31 \pm 0.04$ | $54.59 \pm 0.02$ | $54.54 \pm 0.02$ | $52.94 \pm 0.04$ |
| FedET | | $57.75 \pm 0.01$ | $56.70 \pm 0.02$ | $57.08 \pm 0.01$ | $58.59 \pm 0.01$ |
| FedU/FedEMA | | - | - | - | - |
| Hetero-SSFL | | $\underline{63.20 \pm 0.08}$ | $\underline{63.61 \pm 0.02}$ | $\underline{61.93 \pm 0.04}$ | $\underline{61.15 \pm 0.07}$ |
| FedMKD | | $\mathbf{66.39 \pm 0.09}$ | $\mathbf{68.35 \pm 0.05}$ | $\mathbf{67.60 \pm 0.04}$ | $\mathbf{65.88 \pm 0.03}$ |

Table 7: Top-1 accuracy comparison under linear probing on CIFAR-100 dataset with best model performance in bold and second-best results with underlines. '-' means this method is not suitable for the experiment setting.

| Method | Public dataset | CIFAR-100 (%) | | | |
|---|---|---|---|---|---|
| | | Class | Dir($\beta$=0.1) | Dir($\beta$=0.5) | IID |
| Std. ResNet18 | | $25.35 \pm 0.02$ | | | |
| FedMD | | $23.25 \pm 0.05$ | $23.08 \pm 0.02$ | $22.46 \pm 0.04$ | $23.20 \pm 0.03$ |
| FedDF | | $23.07 \pm 0.03$ | $22.44 \pm 0.06$ | $22.73 \pm 0.03$ | $21.57 \pm 0.01$ |
| MOON-KL | | $21.26 \pm 0.03$ | $20.81 \pm 0.04$ | $21.34 \pm 0.01$ | $21.82 \pm 0.02$ |
| MOON | IID | $27.82 \pm 0.01$ | $27.63 \pm 0.01$ | $26.84 \pm 0.03$ | $26.70 \pm 0.03$ |
| FedET | | $29.11 \pm 0.03$ | $28.55 \pm 0.02$ | $26.98 \pm 0.01$ | $24.48 \pm 0.02$ |
| FedU/FedEMA | | - | - | - | - |
| Hetero-SSFL | | $\underline{30.84 \pm 0.10}$ | $\underline{29.74 \pm 0.03}$ | $\underline{29.63 \pm 0.06}$ | $\underline{28.89 \pm 0.06}$ |
| FedMKD | | $\mathbf{36.33 \pm 0.01}$ | $\mathbf{34.37 \pm 0.05}$ | $\mathbf{35.59 \pm 0.07}$ | $\mathbf{35.94 \pm 0.02}$ |
| Std. ResNet18 | | $24.97 \pm 0.01$ | | | |
| FedMD | | $23.95 \pm 0.05$ | $23.28 \pm 0.02$ | $23.14 \pm 0.03$ | $22.47 \pm 0.01$ |
| FedDF | | $27.21 \pm 0.02$ | $27.83 \pm 0.02$ | $27.31 \pm 0.04$ | $27.05 \pm 0.04$ |
| MOON-KL | | $21.73 \pm 0.01$ | $22.13 \pm 0.02$ | $20.97 \pm 0.03$ | $22.27 \pm 0.04$ |
| MOON | Partial | $27.00 \pm 0.04$ | $27.05 \pm 0.06$ | $27.27 \pm 0.01$ | $28.26 \pm 0.02$ |
| FedET | | $29.38 \pm 0.01$ | $29.56 \pm 0.02$ | $28.12 \pm 0.02$ | $\underline{29.61 \pm 0.01}$ |
| FedU/FedEMA | | - | - | - | - |
| Hetero-SSFL | | $\underline{30.94 \pm 0.05}$ | $\underline{30.78 \pm 0.07}$ | $\underline{29.92 \pm 0.03}$ | $29.56 \pm 0.03$ |
| FedMKD | | $\mathbf{35.82 \pm 0.02}$ | $\mathbf{34.70 \pm 0.02}$ | $\mathbf{35.55 \pm 0.05}$ | $\mathbf{34.38 \pm 0.02}$ |

VGG9. But for *Hetero-SSFL*, each client needs to upload the kernel metric $\mathcal{K} \in \mathbb{R}^{L \times L}$, $L = |D_{pub}|$ is the size of the public dataset. In order to make comparison, we fix the communication rounds. Compared to the *Hetero-SSFL*, when $S(\theta_n) * 2 \leq |D_{pub}|^2$, our *FedMKD* is better in communication cost. And compared to the *MOON*, for larger server model, $S(\theta_n) * 2 \leq S(\theta_s) * 2$, *MOON* will cost more when the server model is larger.

About local storage for each client, we compute the summary of the model and data. For *FedMKD* and *Hetero-SSFL*, each client runs the siamese network, so the model storage is double of the model memory, that is $2 * S(\theta_n)$. But for the *MOON*, each client processes the local contrastive learning and the model contrastive learning, so they must store two local models and one global model, that is $2 * S(\theta_n) + S(\theta_s)$. Here, we neglect the storage of the predictor which is usually a 2-layer MLP. We use $S(|D|)$ to denote the storage of the dataset. And each client in *FedMKD* only needs to store the

Table 8: Top-1 accuracy comparison on 1% of labeled data for semi-supervised learning on CIFAR-10 dataset with best model performance in bold and second-best results with underlines. '-' means this method doesn't apply for the experiment setting.

| Method | Public dataset | CIFAR-10 (%) | | | |
|---|---|---|---|---|---|
| | | **Class** | **Dir($\beta$=0.1)** | **Dir($\beta$=0.5)** | **IID** |
| Std. ResNet18 | | $46.84 \pm 0.25$ | | | |
| FedMD | | $43.32 \pm 0.22$ | $43.83 \pm 0.25$ | $44.20 \pm 0.18$ | $44.66 \pm 0.20$ |
| FedDF | | $43.60 \pm 0.44$ | $43.73 \pm 0.36$ | $44.13 \pm 0.16$ | $44.80 \pm 0.40$ |
| MOON-KL | IID | $45.42 \pm 0.26$ | $46.18 \pm 0.38$ | $46.61 \pm 0.21$ | $46.72 \pm 0.15$ |
| MOON | | $49.96 \pm 0.24$ | $49.90 \pm 0.32$ | $50.21 \pm 0.10$ | $51.78 \pm 0.28$ |
| FedET | | $52.37 \pm 0.24$ | $54.29 \pm 0.25$ | $56.57 \pm 0.17$ | $57.44 \pm 0.13$ |
| FedU/FedEMA | | - | - | - | - |
| Hetero-SSFL | | $\underline{54.30 \pm 0.15}$ | $\underline{56.19 \pm 0.21}$ | $\underline{58.73 \pm 0.54}$ | $\underline{60.50 \pm 0.12}$ |
| FedMKD | | $\mathbf{59.65 \pm 0.28}$ | $\mathbf{60.71 \pm 0.05}$ | $\mathbf{61.78 \pm 0.40}$ | $\mathbf{64.06 \pm 0.32}$ |
| Std. ResNet18 | | $46.42 \pm 0.12$ | | | |
| FedMD | | $44.54 \pm 0.26$ | $44.01 \pm 0.19$ | $43.61 \pm 0.13$ | $42.52 \pm 0.19$ |
| FedDF | | $48.14 \pm 0.27$ | $47.95 \pm 0.17$ | $48.74 \pm 0.12$ | $48.56 \pm 0.18$ |
| MOON-KL | | $46.76 \pm 0.05$ | $47.52 \pm 0.20$ | $46.92 \pm 0.05$ | $46.49 \pm 0.22$ |
| MOON | Partial | $50.43 \pm 0.18$ | $51.06 \pm 0.20$ | $51.99 \pm 0.34$ | $49.86 \pm 0.19$ |
| FedET | | $52.75 \pm 0.07$ | $52.12 \pm 0.10$ | $52.61 \pm 0.03$ | $54.64 \pm 0.14$ |
| FedU/FedEMA | - | - | - | - | - |
| Hetero-SSFL | | $\underline{59.95 \pm 0.34}$ | $\underline{59.34 \pm 0.08}$ | $\underline{58.31 \pm 0.50}$ | $\underline{58.35 \pm 0.20}$ |
| FedMKD | | $\mathbf{61.55 \pm 0.19}$ | $\mathbf{62.99 \pm 0.17}$ | $\mathbf{63.10 \pm 0.28}$ | $\mathbf{61.08 \pm 0.19}$ |

Table 9: Top-1 accuracy comparison on 1% of labeled data for semi-supervised learning on CIFAR-100 dataset with best model performance in bold and second-best results with underlines. '-' means this method doesn't apply for the experiment setting.

| Method | Public dataset | CIFAR-100 (%) | | | |
|---|---|---|---|---|---|
| | | **Class** | **Dir($\beta$=0.1)** | **Dir($\beta$=0.5)** | **IID** |
| Std. ResNet18 | | $15.19 \pm 0.20$ | | | |
| FedMD | | $15.88 \pm 0.17$ | $15.01 \pm 0.14$ | $14.94 \pm 0.19$ | $15.34 \pm 0.12$ |
| FedDF | | $14.39 \pm 0.20$ | $14.22 \pm 0.03$ | $13.06 \pm 0.14$ | $12.90 \pm 0.07$ |
| MOON-KL | | $16.25 \pm 0.06$ | $16.48 \pm 0.13$ | $17.22 \pm 0.25$ | $16.07 \pm 0.04$ |
| MOON | IID | $19.23 \pm 0.12$ | $18.74 \pm 0.13$ | $17.21 \pm 0.13$ | $17.07 \pm 0.18$ |
| FedET | | $19.70 \pm 0.08$ | $17.53 \pm 0.09$ | $16.82 \pm 0.20$ | $15.68 \pm 0.18$ |
| FedU/FedEMA | | - | - | - | - |
| Hetero-SSFL | | $\underline{20.04 \pm 0.40}$ | $\underline{19.05 \pm 0.09}$ | $\underline{19.19 \pm 0.17}$ | $\underline{18.82 \pm 0.17}$ |
| FedMKD | | $\mathbf{22.57 \pm 0.12}$ | $\mathbf{21.28 \pm 0.24}$ | $\mathbf{22.13 \pm 0.11}$ | $\mathbf{22.07 \pm 0.15}$ |
| Std. ResNet18 | | $14.12 \pm 0.09$ | | | |
| FedMD | | $17.32 \pm 0.17$ | $16.91 \pm 0.11$ | $16.47 \pm 0.14$ | $16.31 \pm 0.07$ |
| FedDF | | $17.01 \pm 0.04$ | $17.60 \pm 0.10$ | $17.14 \pm 0.01$ | $16.95 \pm 0.01$ |
| MOON-KL | | $16.21 \pm 0.27$ | $16.87 \pm 0.11$ | $16.10 \pm 0.12$ | $16.94 \pm 0.12$ |
| MOON | Partial | $18.64 \pm 0.21$ | $19.39 \pm 0.28$ | $18.92 \pm 0.25$ | $19.29 \pm 0.08$ |
| FedET | | $18.49 \pm 0.12$ | $19.71 \pm 0.16$ | $18.16 \pm 0.13$ | $18.01 \pm 0.22$ |
| FedU/FedEMA | | - | - | - | - |
| Hetero-SSFL | | $\underline{20.72 \pm 0.14}$ | $\underline{20.83 \pm 0.23}$ | $\underline{20.30 \pm 0.32}$ | $\underline{19.62 \pm 0.08}$ |
| FedMKD | | $\mathbf{22.44 \pm 0.19}$ | $\mathbf{21.45 \pm 0.16}$ | $\mathbf{22.21 \pm 0.17}$ | $\mathbf{20.94 \pm 0.25}$ |

Table 10: Communication and storage cost comparison of FedMKD and several baselines.

| Methods | Communication cost | | Storage cost | |
| --- | --- | --- | --- | --- |
| | Resnet18 | VGG9 | Resnet18 | VGG9 |
| FedMKD | 85.26MB | 32.76MB | 193.07MB | 140.57MB |
| MOON | 85.25MB | | 235.70MB | 183.20MB |
| Hetero-SSFL | 122.08MB | | 239.95MB | 187.45MB |

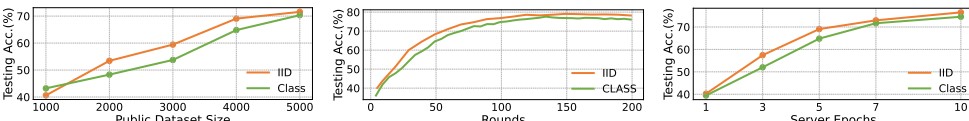

(a) The performance of FedMKD with different public dataset size on Cifar-10.
(b) The performance of FedMKD with different global rounds on Cifar-10.
(c) The performance of FedMKD with different server epochs on Cifar-10.

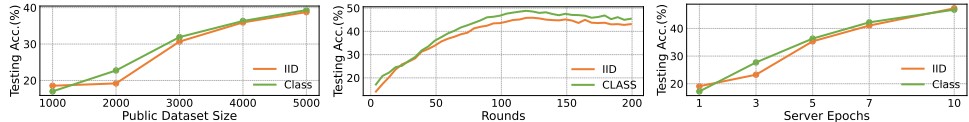

(d) The performance of FedMKD with different public dataset size on Cifar-100.
(e) The performance of FedMKD with different global rounds on Cifar-100.
(f) The performance of FedMKD with different server epochs on Cifar-100.

Figure 7: The top-1 test accuracy of different hyperparameter settings on CIFAR-10 and CIFAR100

local training data without the public data, that is $S(|D_n|)$ and for *Hetero-SSFL* client needs to store both two $S(|D_n|) + S(|D_{pub}|)$. *MOON* doesn't use the public dataset, so the client only needs to store the local data $S(|D_n|)$also. Totally, we can get

$$2 * S(\theta_n) + S(|D|) \leq 2 * S(\theta_n) + S(|D_n|) + S(|D_{pub}|),$$
$$2 * S(\theta_n) + S(|D|) \leq 2 * S(\theta_n) + S(\theta_s) + S(|D|).$$

### D.3 Hyperparameter analysis

To explore the influence of different hyperparameters, we conduct experiments on several key parameters of *FedMKD*.

**Impact of public dataset size.** As mentioned in [20], the size and construct method of the public dataset are both critically important, especially since our global model needs training based on it. About the latter one we've given an analysis in the last section, so we explore the influence of the size on the model performance. Considering the construct method of the public dataset, we choose part of the dataset as a public dataset and set it aside to be used. The performance change with respect to its size is shown in Fig. 7 (a)(d). We can observe that in both two datasets, when the public dataset is small, the performance of *FedMKD* is quite worse, and as the size of the public dataset increases, the performance gets better. But when the size is larger than 4000, the gain of performance is small, and because of the design of the model, the time cost of the server computation is larger, so we choose 4000 as our experiment setting.

**Impact of the number of global rounds.** In order to investigate the impact of the training rounds to *FedMKD*, we fix the other hyperparameters and train the model for 200 rounds. The results showed in Fig. 7 (b)(e) demonstrate that while the total training rounds increase, the performance of the *FedMKD* gets better. For both two kinds clients data distribution in *CIFAR-10*, the global model converges at nearly 150 rounds. For *CIFAR-100*, the global model converges at 200 rounds. There's no denying that the increase of the global rounds will result in better performance, especially when the global round is small, the improvement of the performance is significant.

**Impact of the number of server epochs.** Since our global model is trained on the public dataset, through standalone training and knowledge distillation. The lower server epoch means the global model may under-fitting not only the distribution of the public dataset but also the client knowledge. On the other hand, large server epoch will lead large computation cost, and over-fit the distribution of the public dataset, which is harmful to the generalization of the model, so the epoch in server is important. We choose 5 different settings $T' = \{1, 3, 5, 7, 10\}$ to validate the influence of the server epochs. Based on the experiment results shown in Fig. 7(c)(f), we can see that the performance of the global model improves when the server epoch increases with the distribution of the public is IID. But the change of improvement gets smaller when the server epoch is larger than 5.

## E  Additional discussion

In this section, we discuss the limitations and broader impacts of the work.

**Limitations.** Although we provide detailed explanations of the proposed algorithm and extensive experiments analysis, the theoretical proof of *FedMKD* is not rigorous enough. The characteristic of multi-teacher distillation of the global model has not been sufficiently theoretically justified and we only combined it with the self-training loss as one whole loss to analyze.

**Broader Impacts.** FedMKD offers significant societal and technological benefits, which is crucial in kinds of domains like healthcare and finance. It promotes inclusivity by leveraging diverse data from various sources, thereby reducing biases and improving model generalization. Technologically, FedMKD lowers the dependency on labeled data, making federated self-supervised learning more efficient and scalable, and drives innovation in heterogeneous resource-limited devices. By carefully navigating these challenges: deviated representation abilities and inconsistent representation spaces, FedMKD can lead to responsible and equitable advancements in distributed AI technology.

