# OpenReview forum: "Resource-Aware Federated Self-Supervised Learning with Global Class Representations"
_NeurIPS.cc/2024/Conference — NeurIPS 2024 poster_

### Official Review · Reviewer_hPCk · 2024-07-09

**Soundness:** 3
**Presentation:** 3
**Contribution:** 3
**Rating:** 6
**Confidence:** 4

**Summary:**

This paper introduces a novel approach for enhancing global representation models in resource-adaptive federated self-supervised learning through a multi-teacher knowledge distillation framework, named FedMKD. The proposed method addresses the challenges posed by heterogeneous architectures and extreme class skew, demonstrating significant improvements in representation abilities across diverse clients. The authors provide detailed experimental results to support their claims.

**Strengths:**

1. The paper tackles a critical issue in federated self-supervised learning, offering a unique and effective solution.
2. FedMKD effectively leverages multi-teacher knowledge distillation to integrate knowledge from heterogeneous clients, even under extreme class skew.
3. The adaptive knowledge integration mechanism enhances the representation abilities of heterogeneous models.
4. The experimental section is thorough, demonstrating the efficacy of the proposed method on multiple datasets.
5. The combination of self-supervised loss and distillation loss, along with the global knowledge anchored alignment module, significantly improves local and global representation abilities.

**Weaknesses:**

The Related Work section could be expanded to include a more thorough discussion of existing approaches in both transfer learning and attention mechanisms.
The paper may have a limited audience due to its specialized focus on image classification tasks. It could benefit from providing more context for readers unfamiliar with the specific datasets and techniques discussed.
The contributions of the work could be more explicitly stated in the Introduction to enhance clarity for readers.

**Questions:**

It would be beneficial for the authors to include a complexity analysis of the FedMKD algorithm to provide insights into its computational efficiency. If a complexity analysis is not feasible, the authors should provide a rationale for its omission.

**Limitations:**

The limitations of the FedMKD framework are well-addressed in the paper. These constraints appear to be inherent to the design choices made in the method and are necessary trade-offs for its functionality.

---

> ### Author Rebuttal · Authors · 2024-08-07
>
> We thank the reviewer hPCk for the time and valuable feedback! We would try our best to address the comments one by one.
>
> **Response to Weakness:**
>
> Thank you very much for the insightful comments.
>
> - First, we survey some **federated transfer learning (FTL)** methods, and add the following discussion of them.
> "Federated transfer learning aims to share the knowledge and insights derived from training machine learning models across various entities, all while keeping the raw data decentralized. It has been used widely in various regions, such as cross-domain recommendation [1,2], medical image classification [3,4] and financial service [5,6]"
> About **attention mechanisms**, we would add the following discussion as follows.
> "Attention mechanism is a component used in neural networks that dynamically focuses on the most relevant parts of the input data, enhancing the model's performance. By assigning varying levels of importance to different pieces of information, it allows the network to prioritize and efficiently process significant elements, has proven to be highly successful in various tasks, such as image classification [7,8], object detection [9,10] and image generation [11,12]."
> Both of these discussions will be added in Related Work in the final version.
>
> - By the way, we would add more detailed description for readers who are unfamiliar with the specific datasets and techniques to understand our method as follows. "We use CIFAR-10 and CIFAR-100 to evaluate the performance. The CIFAR-10 dataset consists of 60000 32x32 colour images in 10 classes, with 6000 images per class. There are 50000 training images and 10000 test images. CIFAR-100 dataset is just like the CIFAR-10, except it has 100 classes containing 600 images each. There are 500 training images and 100 testing images per class."
>
>
> - Finally, we present our contributions in **Lines 65-76** of the Introduction. We emphasize that in resource-adaptive Fed-SSL, we are the first to delve into global class representations learning by addressing the deviated representation abilities and inconsistent representations caused by heterogeneous architectures and class skew. We designed **FedMKD** to tackle these challenges, and the experimental results demonstrate the superiority of our proposed **FedMKD**.
>
>
> **Response to Question:**
>
> Thank you very much for the insightful comments. We would like to analyze the complexity of client and server separately. First in the client, each client chooses the appropriate model according to their resource, so the computation complexity is different among them. Then in server, the training process has two phases. The complexity of knowledge distillation is $O(T' \times |D_{pub}| \times (N+1))$, where $T'$ is the epoch of knowledge distillation, $|D_n|$ is the size of the public dataset, and $N$ is the number of client local model. The complexity of the global knowledge anchored is $O(|D_{pub}| \times N)$. By the way, considering that each deep network includes multiple operations and computations, we commonly use the amount of computation (FLOPS) to analyze the time complexity and the amount of memory access (Bytes) to analyze the space complexity. The results of each process are shown as follows:
> |  **Process** 	| **Memory** |**FLOPs** |
> |:------------:|:-------:|:-------:|
> | Local-VGG9 | 37.857M |121.761G|
> | Local-ResNet18| 49.294M |440.813G|
> | Distillation | 343.931M |1.701T|
> | Alignment | 60.129M |295.309G|
>
>
> If there are any further confusions/questions, we are happy to clarify and try to address them. Thank you again and your recognition means a lot for our work.
>
> ---
> [1] Ammad-Ud-Din, Muhammad, et al. "Federated collaborative filtering for privacy-preserving personalized recommendation system." arXiv preprint arXiv:1901.09888 (2019).
>
> [2] Minto, Lorenzo, et al. "Stronger privacy for federated collaborative filtering with implicit feedback." Proceedings of the 15th ACM Conference on Recommender Systems. 2021.
>
> [3] Gong, Xuan, et al. "Federated learning with privacy-preserving ensemble attention distillation." IEEE transactions on medical imaging 42.7 (2022): 2057-2067.
>
> [4] Sui, Dianbo, et al. "Feded: Federated learning via ensemble distillation for medical relation extraction." Proceedings of the 2020 conference on empirical methods in natural language processing (EMNLP). 2020.
>
> [5] Shaheen, Momina, Muhammad Shoaib Farooq, and Tariq Umer. "Reduction in data imbalance for client-side training in federated learning for the prediction of stock market prices." Journal of Sensor and Actuator Networks 13.1 (2023): 1.
>
> [6] Pourroostaei Ardakani, Saeid, et al. "A federated learning-enabled predictive analysis to forecast stock market trends." Journal of Ambient Intelligence and Humanized Computing 14.4 (2023): 4529-4535.
>
> [7] Hu, Jie, Li Shen, and Gang Sun. "Squeeze-and-excitation networks." Proceedings of the IEEE conference on computer vision and pattern recognition. 2018.
>
> [8] Woo, Sanghyun, et al. "Cbam: Convolutional block attention module." Proceedings of the European conference on computer vision (ECCV). 2018.
>
> [9] Dai, Jifeng, et al. "Deformable convolutional networks." Proceedings of the IEEE international conference on computer vision. 2017.
>
> [10] Carion, Nicolas, et al. "End-to-end object detection with transformers." European conference on computer vision. Cham: Springer International Publishing, 2020.
>
> [11] Gregor, Karol, et al. "Draw: A recurrent neural network for image generation." International conference on machine learning. PMLR, 2015.
>
> [12] Zhang, Han, et al. "Self-attention generative adversarial networks." International conference on machine learning. PMLR, 2019.

---

> > ### Comment · Reviewer_hPCk · 2024-08-13
> >
> > After reviewing the responses and other reviewers' comments, I appreciate the detailed clarifications and new experiments provided. The explanations effectively addressed my concerns, leading me to update my rating to weak accept.

---

> > > ### Author Response · Authors · 2024-08-13
> > >
> > > Thank you very much for taking the time to read and respond to our rebuttal. We sincerely appreciate your recognition of our work.

---

### Official Review · Reviewer_sGaU · 2024-07-10

**Soundness:** 3
**Presentation:** 3
**Contribution:** 4
**Rating:** 8
**Confidence:** 5

**Summary:**

The authors propose a multi-teacher knowledge transfer framework, FedMKD, based on two challenges in resource-adaptive federated self-supervised learning: deviated representations abilities and inconsistent representations. This framework uses an adaptive knowledge integration mechanism and a weighted combination of different loss functions to ensure that global representations with class-wide knowledge can be learned from heterogeneous clients even in the presence of extreme class skew. Extensive experiments on two datasets demonstrate the effectiveness of FedMKD, which outperforms the state-of-the-art baseline by an average of 4.78% under linear evaluation.

**Strengths:**

1.	The scenarios presented and the challenges solved in this work are of importance in reality and the proposed multi-teacher knowledge distillation based federated self-supervised learning framework is able to solve the challenges effectively.

2.	By innovatively combining self-supervised loss and distillation loss, FedMKD can encode skew classes into a unified space, which is not covered by related work.

3.	The paper is easy to follow, including the main paper and appendices. The introduction provides a clear motivation and a brief overview of the proposed framework, and the other sections provide detailed descriptions of the method.

4.	The experiments are well-organized and successful, making this work more convincing.

**Weaknesses:**

1.	The mechanism design seems to be confusing. As shown in the FedMKD, multi-instructor knowledge distillation is allowing global models to learn local models, while global knowledge anchored alignment is allowing local models to learn global models, knowledge transfer happens alternately.

2.	FedMKD aims to learn global class representations according to a global model. But an additional global-anchored mechanism improves the local performance. What’s the difference between personalized federated learning and the proposed FedMKD?

3.	It’s suggested that the authors could add the instructions about the whole model training algorithm in Appendix A, which can help reviewers understand the whole process.

4.	The presentation needs to be improved, especially for some equations. The self-supervised loss value in Eq. (10) was not discussed earlier. If it is equal to the loss value in Eq.(3), but with a different notation.

**Questions:**

1.	As shown in Weakness 1, it seems to that multi-instructor knowledge distillation is allowing global models to learn local models, while global knowledge anchored alignment is allowing local models to learn global models, so does the two mechanisms result in duplication of knowledge transfer?

2.	In the problem statement, the main objective is the global loss. Since recent studies on cross-device scenarios also consider personalized FL (PFL), how about the personalization? Can your method be extended to PFL?

**Limitations:**

The authors pointed out the limitations of the paper that is the theoretical proof of FedMKD is not rigorous enough, the characteristic of multi-teacher distillation of the global model has not been sufficiently theoretically justified.

---

> ### Author Rebuttal · Authors · 2024-08-07
>
> We thank the reviewer sGaU for the time and valuable feedback! We would try our best to address the comments one by one.
>
> **Response to Weakness1 & Question1:**
>
> Thank you very much for your recognition. As you mentioned, we have two mechanisms to transfer knowledge, but they are not duplicated. First, the multi-teacher knowledge distillation allows the global model to encode all classes from clients in a unified space. And the global knowledge-anchored alignment module is applied to eliminate inconsistency in representation spaces of clients, which is benefit to the global model training, also. These two machenisms are critical for learning global class representation and are indispensable.
>
> **Response to Weakness2 & Question2:**
>
> Thank you for your insightful comments. The additional global anchored mechanism aims to ensure that the local representation spaces are closer to the global ones so that the following knowledge distillation process could gain more from clients. This module can further eliminate the effect of inconsistent representation spaces and improve local performance. The experiment result of the improvement in client is shown in Section 5.4 which verifies the effectiveness of the module. Compared to personalized FL, the optimization objective is not identical. We aim to train a global model that can encode all classes in a unified space, and the improvement of clients is a middle process. But for PFL, it aims to learn better local model without global model, it's quite different from ours.
>
> **Response to Weakness3:**
>
> As you suggested, we would add the instructions about the whole model training algorithm in Appendix A in our final version.
>
> **Response to Weakness4:**
>
> We are sorry for our unclear discussion. Eq. (10) shows the weighted combined loss for the global model, it includes two parts, the self-supervised loss and knowledge distillation loss. So that the global model can encode all classes from clients in a unified space and improve its representation capability. The self-supervised one is identical to the local training loss, that is Eq. (3).
>
> If there are any further confusions/questions, we are happy to clarify and try to address them. Thank you again and your recognition means a lot for our work.

---

> > ### Comment · Reviewer_sGaU · 2024-08-09
> >
> > Thank you for your response ! I appreciate the thoughtful consideration given to my comments. The explanations and revisions provided are clear and effectively address my concerns, and I strongly recommend the acceptance of this paper.

---

> > > ### Author Response · Authors · 2024-08-12
> > > **Response to Reviewer sGaU**
> > >
> > > We appreciate you taking time to read and respond to our rebuttal. And thank you very much again for recognizing our work.

---

### Official Review · Reviewer_24fB · 2024-07-12

**Soundness:** 3
**Presentation:** 3
**Contribution:** 2
**Rating:** 5
**Confidence:** 4

**Summary:**

This paper proposes a multi-teacher knowledge distillation framework named FedMKD for resource-adaptive federated self-supervised learning (Fed-SSL). The method aims to address the challenges of global representation learning in Fed-SSL caused by heterogeneous architectures and imbalanced class distributions. Experimental results demonstrate that FedMKD is significantly effective in addressing heterogeneity and class distribution imbalance issues in federated self-supervised learning.

**Strengths:**

Innovative Framework Design: This paper proposes a new multi-teacher knowledge distillation framework, FedMKD, which combines self-supervised learning and knowledge distillation to effectively address the challenges of heterogeneity and data imbalance in federated learning.

Adaptive Knowledge Integration Mechanism: Through an adaptive weighting strategy, it integrates high-quality representations from heterogeneous models, significantly enhancing the representation capabilities of the global model.

Global Knowledge Anchored Alignment: The design of a global knowledge anchored alignment module ensures that local models' representation spaces are closer to the global representation space, thus improving the performance of local models.

**Weaknesses:**

1. This paper combines federated self-supervised learning and resource-aware federated learning. However, in introduction and related works, the authors lack a discussion about some works about resource-aware federated learning with heterogeneous clients.
.
2. The authors assume to build a global dataset to perform knowledge distillation and anchored alignment. However, random sampling makes the global dataset and client dataset have the similar data distribution, which violates the original intention of data privacy in FL. Please provide some explanation for this.

3. CIFAR-10 and CIFAR-100 are both image datasets with relatively small sizes and low image resolution, which cannot fully reflect the complexity of real-world applications. It is recommended to add experiments on large-scale image datasets such as Tiny-ImageNet and ImageNet-100.

4. The framework of dual encoders increases the training load in clients. It is recommended to evaluate the computation time and resources required and provide analysis for local model training.

5. To evaluate the robustness and effectiveness of this framework, it is suggested to explore experiment settings with more clients, such as [1].

[1] ScaleFL: Resource-Adaptive Federated Learning with Heterogeneous Clients

**Questions:**

See weakness.

**Limitations:**

The characteristic of multi-teacher distillation of the global model has not been sufficiently theoretically justified and the authors only combined it with the self-training loss as one whole loss to analyze.

---

> ### Author Rebuttal · Authors · 2024-08-07
>
> We thank the reviewer 24fB for the time and valuable feedback! We would try our best to address the comments one by one.
>
> **Response to Weakness 1:**
>
> Thank you very much for the insightful comments. In the related works section (Lines 94-99), we have surveyed several existing studies that address heterogeneous client models in federated self-supervised learning. We analyzed the advantages and disadvantages of **Hetero-SSFL** and **FedFoA** in detail. Additionally, we select **Hetero-SSFL** as one of the baselines in the experiment to compare and evaluate the performance of our proposed method. And as you mentioned, we will further add the description of **Hetero-SSFL** and **FedFoA** in the introduction in the final version.
>
> **Response to Weakness 2:**
>
> Thanks for your sincere comments.
> As you are concerned, this construction method of public dataset is the experimental simulation part only does not pose a serious risk of data leakage, in our FedMKD, no data need to be transferred. Here, we just want to address one practical problem: if the distribution of public dataset is different from the data distributions of clients, can the proposed FedMKD still work?
>
> In practice we can use appropriate
> public dataset to distill knowledge according to the specific task instead of this random sampling method. To verify this, we conducted another experiment using **CINIC** as the public dataset on the server. **CINIC** is an extension of CIFAR-10, augmented with downsampled ImageNet images, and it shares the same size and classes as CIFAR-10.
> We still use **CIFAR-10** as the train dataset for the clients. The public dataset is 'Partial' and the data distribution in the client is 'Class'. The other setting is identical to the draft. The linear probing results are shown as follows:
>
> | |  **Public dataset** | **Acc**|
> |:--:|:--:|:---:|
> |FedMKD| CINIC | 59.98% |
> |FedMKD| CIFAR-10 | 66.39% |
>
> Here we can see that even when intuitively choosing an appropriate public dataset, our method still works effectively.
>
> **Response to Weakness 3:**
>
> Thank you very much for the insightful comments. We use **ImageNet-100** and **ImageNet-1k** to evaluate the effectiveness of our proposed method. The results are presented in **Table 11 in Global Response PDF**. We can conclude that our proposed FedMKD outperforms all baselines.
>
> **Response to Weakness 4:**
>
> Thank you very much for the insightful comments. We evaluate another single-encoder framework, **SimCLR**, on the clients and assessed the computation time and resources required for local model training. Although this method is more computationally efficient, its performance is much inferior to ours. The results are shown as follows:
>
> | |  **Local method** 	| **Acc**| **VGG Memory** |**ResNet18 Memory**|
> |:--:|:--:|:---:|:---:|:---:|
> |FedMKD| SimCLR-single encoder| 48.32% | 21.066M|21.668M|
> |FedMKD| BYOL-dual encoder| 66.39% | 37.857M|49.294M|
>
> **Response to Weakness 5:**
>
> Thank you very much for the insightful comments. We've explored the scalability of our proposed algorithm FedMKD in Appendix D.3. In these experiments, we test scenarios with 10 and 30 clients, where 40% of the clients use the VGG model and 60% use ResNet18. We repartition the data for each client under Dir($\beta=0.5$) and set the public dataset distribution as IID. To compare scalability performance, we conduct the same experiments on the second-best baseline, Hetero-SSFL. The linear probing results are shown as follows:
>
> |  **Method** 	| **5-Client**| **10-Client** |**30-Client** |
> |:------------:|:-------:|:-------:|:-------:|
> | Hetero-SSFL | 59.13% | 54.51% | 49.22% |
> | Ours| 67.79% | 57.39% | 53.85%|
>
> The results show that as the number of clients increases, our proposed method still outperforms **Hetero-SSFL**, demonstrating the scalability of our **FedMKD**.
>
> If there are any further confusions/questions, we are happy to clarify and try to address them. Thank you again and your recognition means a lot for our work.

---

### Official Review · Reviewer_de59 · 2024-07-13

**Soundness:** 3
**Presentation:** 3
**Contribution:** 2
**Rating:** 4
**Confidence:** 4

**Summary:**

This paper studies the problem of federated self-supervised learning. A multi-teacher knowledge distillation framework is proposed to address the two challenges: deviated representation abilities and inconsistent representation spaces. Specifically, the adaptive knowledge integration mechanism is designed to learn better representations from all heterogeneous models, and a global knowledge-anchored alignment module is used to make the local representation spaces close to the global spaces. Experiments on two datasets demonstrate the effectiveness of the proposed method.

**Strengths:**

* This paper is well-written and easy to follow.
* The analysis of the challenges, i.e., deviated representation abilities and inconsistent representation spaces, is clear.
* Introducing an attention module to integrate the knowledge is technique sound.

**Weaknesses:**

* Knowledge distillation is widely used in federated learning and federated self-supervised learning. The contribution of the proposed method (i.e., multi-teacher knowledge distillation framework) is incremental.
* This paper uses small models such as resnet18 and vgg9.
* Only CIFAR datasets are used to evaluate the proposed method. Experiments on ImageNet-1k are important to evaluate the effectiveness.

**Questions:**

Weakness.

**Limitations:**

The authors adequately addressed the limitations in the appendix.

---

> ### Author Rebuttal · Authors · 2024-08-07
>
> We thank the reviewer de59 for the time and valuable feedback! We would try our best to address the comments one by one.
>
> **Response to Weakness 1:**
>
> Thank you very much for the insightful comments.
> As you said, knowledge distillation (KD) has been widely used in Fed-SSL, such as FedX and other works. However, our contrastive learning-based multi-teacher KD introduces novel approaches to address the challenges of **deviated representation abilities** and **inconsistent representation spaces**:
> - First, to mitigate the impact of heterogeneous models with **deviated representation abilities**, we introduce an **adaptive knowledge integration** module to learn high-quality representations from them.
> - Second, to encode all classes from clients in a unified space, the global model updates using a **weighted combination of self-supervised loss and distillation loss** to update.
> - Finally, the global knowledge anchored alignment module is applied on the server to eliminate the **inconsistency in representation spaces** and reduce the burden on the clients.
>
> In our experiments, FedMD and FedDF, which are based on traditional KD methods, were adopted as comparative baselines. As shown in Table 2 (Page 7) and Table 3 (Page 8), our proposed FedMKD achieves the best performance.
>
>
> **Response to Weakness 2:**
>
> Thank you very much for you valuable comment.
> Adopting larger models can indeed demonstrate the scalability of our proposed FedMKD framework.
> As suggested, we set one client's local model to **ResNet50** and re-conduct the experiment under the setting of the 'Partial' public dataset and 'Class' distribution among clients on **CIFAR-10**. All other settings remained identical to those in the original paper. The linear probing results are shown below:
>
> |  **Method** 	| **1 ResNet50**|**Original**|
> |:---:|:---:|:-:|
> | FedMD| 48.37% |47.16%|
> | FedDF| 54.46% |52.59%|
> | MOON-KL| 48.48% |46.41%|
> | MOON| 56.33% |54.31%|
> | FedET| 58.96% |57.75%|
> | Hetero-SSFL| 64.74% |63.20%|
> | FedMKD| 67.75% |66.39%|
>
> The results indicate that the introduction of a larger model can improve overall performance, and our proposed FedMKD still outperform all baselines.
>
> **Response to Weakness 3:**
>
> Thank you very much for the insightful comments. We use **Imagenet-100** and **Imagenet-1k** to evaluate the effectiveness of our proposed method. The results are presented in **Table 11 in Global Response PDF**. We can conclude that our proposed FedMKD outperforms all baselines.
>
> If there are any further confusions/questions, we are happy to clarify and try to address them. Thank you again and your recognition means a lot for our work.

---

### Official Review · Reviewer_QsUu · 2024-07-14

**Soundness:** 3
**Presentation:** 3
**Contribution:** 2
**Rating:** 6
**Confidence:** 3

**Summary:**

- The paper proposes a framework called FedMKD which is a multi-teacher knowledge distillation framework for federated learning.
- To allow for different clients to have different resources, a resource adaptive approach is designed. The approach handles class skew and different architectures.
- The knowledge distillation framework allows for learning representations from the different client representations. An SSL loss is used in addition on the common representation space.
- Different from prior works, a global representation model is learnt despite differences in clients.
- Approach is shown to outperform prior works on CIFAR-10 and 100 datasets.

**Strengths:**

- The paper tackles an important problem in federated learning: heterogeneity in clients both from the perspective of resources and data.
- A simple and effective solution is proposed to tackle to learn a common representation space by using techniques in self-supervised learning knowledge distillation.
- The approach is shown to outperform prior work on CIFAR-10 and 100 benchmarks
- The paper is easy to follow. Ablations and experiments seem thorough (see weaknesses).
- While I have not tried it, authors have provided an implementation of their approach.

**Weaknesses:**

Weaknesses/Concerns & Questions:
- How does the approach compare to the traditional FL settings when aggregation like L107 is done (assuming all client architectures are the same) ?
- Did the authors experiment with applying the knowledge distillation to individual projected features instead of the aggregate ?
- L167: Are the negatives also after the aggregation step ? or from individual projection layers ?
- Additional larger-scale datasets will be helpful to understand whether the approach will scale.

Presentation and writing:
- Table 1 needs more details to be helpful. A more detailed caption should help. More subjective columns like "inconsistent representation space", "deviated representation ability" are not apt for a table like this.

**Questions:**

Please refer to the weaknesses section.

**Limitations:**

The paper has adequately discussed limitations and potential negative impacts.

---

> ### Author Rebuttal · Authors · 2024-08-07
>
> We thank the reviewer QsUu for the time and valuable feedback! We would try our best to address the comments one by one.
>
> **Response to Weakness 1:**
>
> Thank you very much for the insightful comments. Our proposed FedMKD is designed for resource-aware settings, allowing each client to choose an appropriate model to train, even if the model architectures differ among clients. In this context, traditional aggregation methods are ineffective. However, if all clients use the same architecture, the model with the minimal parameters should be chosen.
>
> In the newly designed experiments, both the clients' models and the global model are set to VGG9. The linear probing results compared to FedAVG are as follows:
> |  **Methods** 	| **Acc**|
> |:--:|:--:|
> | FedAVG| 48.32% |
> | **FedMKD**| 66.39% |
>
> In our FedMKD approach, each client can adaptively choose the appropriate local model to fully utilize its computational resources. For example, a client with ample computational resources can select a larger model like ResNet18, while another client with limited resources can opt for a smaller model like VGG9. This flexibility enables FedMKD to outperform FedAVG on the VGG9 model.
>
> **Response to Weakness 2 & 3**
>
> First, we apologize for any confusion caused by our previous presentation. To clarify, in Line 167, the negative sample is derived from the individual projection layers of the global model. In knowledge distillation (KD), we have an aggregated anchor representation $\bar{r_i}$, a positive representation $r_{s,i}$ from global model, and multiple negative representations $r_{s,j}$. The purpose of knowledge distillation is to push the anchor representation $\bar{r_i}$ and the positive representation $r_{s,i}$ closer together in the feature space while pulling the anchor away from the negative representations. Thus, the global model could encode all classes from clients in a unified space. Formally, we define this KD loss function, which is presented in the original paper on Line 168:
>
> $L_{distill}=-log\frac{ exp(sim(r _ {s,i}, \bar{r} _ i)/\tau)}{exp(sim(r _ {s,i},\bar{r} _ i)/\tau)+\sum\limits _ {j\neq i}exp(sim(r _ {s,i},r _ {s,j})/\tau)}.$
>
> We also illustrate this process in the **Global Response PDF (see Figure 8)**.
>
> **Response to Weakness 4:**
>
> Thank you very much for the insightful comments. We use **ImageNet-100** and **ImageNet-1k** to evaluate the effectiveness of our proposed method. The results, presented in **Table 11 of the Global Response PDF**, demonstrate that our proposed FedMKD outperforms all baseline methods.
>
> **About Presentation and writing:**
>
> Thank you very much for the insightful comments. We will revise the caption of Table 1 to "Comparison of federated self-supervised learning methods. √ indicates that the proposed method focuses on this challenge and × indicates that it does not.". Additionally, the columns labeled "inconsistent representation space" and "deviated representation ability" will be replaced with the objective description "data heterogeneity." The modified table is presented as **Table 12 in the Global Response PDF**.
>
> If there are any further confusions/questions, we are happy to clarify and try to address them. Thank you again and your recognition means a lot for our work.

---

> > ### Comment · Reviewer_QsUu · 2024-08-11
> > **Thanks for the responses**
> >
> > I have read the responses and other reviews. Thanks for the detailed detailed clarifications and new experiments. I have updated my rating for the paper.

---

> > > ### Author Response · Authors · 2024-08-12
> > > **Response to Reviewer QsUu**
> > >
> > > We appreciate you taking time to read and respond to our rebuttal. And thank you very much again for recognizing our work.

---

### Author Rebuttal · Authors · 2024-08-07

We thank all the reviewers' valuable comments and feedback, which are great helpful in improving the quality of this paper. We try our best to address the concerns including making preliminary experiments as long as time allows. As the **reviewer** **QsUu**, **de59**, **24fB** concerned, we try our best to conduct **new experiments** on **large-scale image datasetes, ImageNet-100 and ImageNet-1k**. We utilize 70 RTX 3090 for more than 100 hours to conduct these experiments. Due to the time limitation, here we only present preliminary results and more results will be presented in the final version. The setting is as follows: the public dataset is 'Partial' and the data distribution in the client is 'Class'. The other setting is identical to the draft. The linear probing results are shown in **Table 11** in PDF.

**Figure 8** in PDF illustrates the multi-teacher knowledge distillation method in response to **reviewer** **QsUu**'s Weakness 2 and Weakness 3.

**Table 12** in PDF is an updated version of the original Table 1, addressing the feedback of "Presentation and writing" from **reviewer** **QsUu**.

Thank you again for the detailed valuable comments to help improve the quality of our work.

---

### Decision · Program_Chairs · 2024-09-25

**Decision:**

Accept (poster)

**Comment:**

The paper proposes a novel approach, which is a multi-teacher knowledge distillation framework for federated learning. In general, the reviewers are satisfied with the manuscript and the author responses. They have appreciated the proposed framework, writing and experiments. Though one reviewer response is still negative, I feel that the author's response adequately addresses the concerns regarding using other models and additional datasets. The recommendation is to accept the paper. The authors are strongly encouraged to address any remaining concerns in the camera-ready version.